

# A revised terminology for male genitalia in Hymenoptera (Insecta), with a special emphasis on Ichneumonoidea

Davide Dal Pos[1], István Mikó[2], Elijah J. Talamas[3], Lars Vilhelmsen[4] and Barbara J. Sharanowski[1]

[1] Department of Biology, University of Central Florida, Orlando, United States of America
[2] Don Chandler Entomological Collection, University of New Hampshire, Durham, NH, United States of America
[3] Division of Plant Industry, Florida Department of Agriculture and Consumer Services, Gainesville, FL, United States of America
[4] Natural History Museum of Denmark, SCIENCE, University of Copenhagen, Copenhagen, Denmark

## ABSTRACT

Applying consistent terminology for morphological traits across different taxa is a highly pertinent task in the study of morphology and evolution. Different terminologies for the same traits can generate bias in phylogeny and prevent correct homology assessments. This situation is exacerbated in the male genitalia of Hymenoptera, and specifically in Ichneumonoidea, in which the terminology is not standardized and has not been fully aligned with the rest of Hymenoptera. In the current contribution, we review the terms used to describe the skeletal features of the male genitalia in Hymenoptera, and provide a list of authors associated with previously used terminology. We propose a unified terminology for the male genitalia that can be utilized across the order and a list of recommended terms. Further, we review and discuss the genital musculature for the superfamily Ichneumonoidea based on previous literature and novel observations and align the terms used for muscles across the literature.

Corresponding author
Davide Dal Pos, daveliga@gmail.com

# INTRODUCTION

## The importance of a unified morphological terminology

The study of morphology entails the interpretation of anatomical structures shaped by evolutionary processes and their translation into rigorous and consistent data (*Boudinot, 2019*). This practice requires the application of terms and concepts for effectively identifying and describing the structures. However, terminologies employed in different groups of organisms can overlap and cause confusion. Homonyms, identical terms used for non-homologous structures, are widespread and employed to describe structures in unrelated taxa (*e.g.*, *Costa et al., 2013*; *Donkelaar et al., 2017*). At the same time, homologous anatomical traits in related groups of organisms often have inconsistent terminologies (*e.g.*, *Schulmeister, 2001*).

Inconsistent terminologies are widespread within and among insect orders (*Yoder et al., 2010*; *Costa et al., 2013*; *Wirkner et al., 2017*). *Girón et al. (2023)* identified five main reasons for the emergence of inconsistent terminologies within insects, namely: (1) borrowing terms from the vertebrate anatomy (*e.g.*, wings, head); (2) creating terms *de novo* (*e.g.*, sclerite, sternite); (3) applying terms to different insect lineages to refer to similar structures located in similar areas of the body (*e.g.*, *cercus* in Diplura and *cercus* in Hymenoptera); (4) changes in phylogenetic classification that caused a reassessment of the morphological terminology; and (5) deviation in the original application of a term due a subsequent misinterpretation (*e.g.*, the concepts of *volsella*). As pointed out by *Yoder et al. (2010)*, the consequences of having disparate terminologies can negatively impact the comparison of gene expression patterns, comparative morphological and phylogenetic studies, analyses of phenotype variability, integration of descriptive taxonomy and phenomics, and machine learning algorithms.

Recently, efforts to address terminological inconsistencies have been undertaken with the intent to unify the terminology and standardize morphological data, and establish a comparability and communicability framework (*e.g.*, *Vogt, Bartolomaeus & Giribet, 2010*). The idea is to use ontology, the logical and linguistic machinery for interpreting physical observations, to relate conceptual objects defined by their general identities and their specific properties, and to use controlled vocabularies for communication among scientists via stabilization of terminologies (*Deans, Yoder & Balhoff, 2012*; *Boudinot, 2019*; *Girón et al., 2023*). Examples of successful attempts are the Drosophila Anatomy Ontology (DAO) (*Costa et al., 2013*), the Ontology of Arthropod Circulatory Systems (OArCS) (*Wirkner et al., 2017*), the Hymenoptera Anatomy Ontology (HAO) (*Yoder et al., 2010*), and more recently the Insect Anatomy Ontology (*Girón et al., 2023*). Of these, the HAO provides an essential tool for hymenopterists but still lacks terms and concepts employed in different taxonomic groups (*Boudinot, 2018*; *Lanes et al., 2020*; *de Brito, Lanes & Azevedo, 2021*). For instance, within the hyperdiverse superfamily Ichneumonoidea ontological alignments for the different morphological structures are still severely lacking.

Among hexapods, the study of male genitalia has long captivated entomologists due to their essential function, diverse morphology, and mechanical adaptations. Even though in some insect orders, male genitalia play a fundamental role in phylogenetic and taxonomic studies (*e.g.*, *Song & Bucheli, 2010*; *Van Dam, 2014*; *Bollino, Uliana & Sabatinelli, 2018*; *Lackner & Tarasov, 2019*), in Hymenoptera they have been relatively little explored (*e.g.*, *Schulmeister, 2001*; *Schulmeister, 2003*; *Boudinot, 2013*), despite being recognized as a critical source of discrete and size-independent characters for both phylogenetic and taxonomic studies (*e.g.*, *Tuxen, 1970*; *Mikó et al., 2013*; *Boudinot, 2015*). This is surprising given the high diversity and the great variety of forms and ecological roles of the order (more than 150,000 described species) (*Klopfstein et al., 2013*; *Branstetter et al., 2018*). One of the possible causes for this is that these characters tend to suffer from rampant terminological inconsistency (*e.g.*, *Tuxen, 1970*), making study difficult.

### Hymenopteran male genitalia: a terminological nightmare

The external male genitalia can offer many potential characters for taxonomic and phylogenetic studies due to their complexity, variability, and accessibility (at least compared to their internal counterparts). However, there are terminological inconsistencies across studies, likely resulting from two different interpretations of homologies based on two competing theories (*Michener, 1956*). The first, the periphallic origin theory, postulates that the male genitalia derived from true appendicular structures and are homologous across most insect orders (*e.g., Crampton, 1919*; *Peck, 1937a*; *Peck, 1937b*; *Michener, 1944*). This theory was recently corroborated by *Boudinot (2018)* (discussed further below). The second, the phallic origin theory, postulates that at least in Hymenoptera, the male genitalia have arisen *de novo* (*e.g., Snodgrass, 1935*; *Snodgrass, 1941*; *Snodgrass, 1957*). Early studies (*e.g., Boulangé, 1924*; *Beck, 1933*; *Snodgrass, 1941*; *Michener, 1956*; *Smith, 1969*; *Smith, 1970a*; *Smith, 1970b*; *Togashi, 1970*; *Smith, 1972*; *Birket-Smith, 1981*; *Kopelke, 1981*) attempted to provide a list of synonymous terms, but suffered from mistakes and incongruities, leading to an increase, rather than a reduction, of the confusion.

It was only with *Schulmeister (2001)* that the first modern and comprehensive work was produced, combining a complete list of synonyms with an attempt to understand the organization of the male copulatory organs in the basal lineages of Hymenoptera. *Schulmeister (2003)* then extended the analysis to the other Hymenoptera, providing the first morphological matrix for the order, making extensive use of characters from the male genitalia. *Schulmeister*'s (*2001*, *2003*) analyses facilitated and bolstered further studies of genitalia within Hymenoptera, allowing subsequent refinement of terms (*e.g., Boudinot, 2013*; *Mikó et al., 2013*). However, more recently, *Boudinot (2018)* rejected the phallic origin theory and provided a new genital terminology for basal Hymenoptera, generating more confusion.

To ameliorate this terminological quagmire, and to facilitate future taxonomic and evolutionary studies, a modern study of the hymenopteran male genitalia is hereby presented. The current contribution provides: (1) a thorough review of the literature; (2) a list of the preferred terminology for the different skeleto-muscular elements of Hymenoptera accompanied by a list of synonyms; (3) the first unified terminology for the skeleto-muscular element of Ichneumonoidea; (4) an alignment of the musculature across the order Hymenoptera; and (5) confirmation of the presence or absence of muscles within Ichneumonoidea.

## MATERIALS AND METHODS

### Sample preparations and imaging

Specimens used for dissection and imaging via confocal laser scanning microscope (CLSM) were collected in Manitoba (Canada), Florida (USA) and Arizona (USA) (Table 1), preserved in 70–90% ethanol and deposited at the University of Central Florida Collection (UCFC). Specimens collection at the Hal Scott Regional Preserve and Park was approved by St. Johns River Water Management District. Male genitalia were dissected by means of minuten pins under a dissecting stereomicroscope OPTIKA SZM-2 which was also used

for observations. Specimens used for CLSM imaging were bleached in 30% H2O2 for two hours, then placed in a droplet of glycerol and imaged with a ZEIS 710 CLSM at the microscope facility of the Burnett School of Biomedical Sciences (University of Central Florida) using 405 and 488nm lasers (following *Mikó & Deans (2013)*). Autofluorescence was collected using three channels with assigned contrasting pseudocolors (420–520 nm, blue; 490–520 nm, green; and 570–670 nm, red). Volume-rendered images and media files were generated using ImageJ (*Schindelin et al., 2015*). Male genitalia not used in CLSM imaging were left to dry and then glued to the tip of a minuten pin and imaged using a Canon Eos 7D camera with a Canon MP-E 65 mm f/2.8 1-5 ×Macro and an M Plan Apo 10×Mitutoyo objective mounted onto the EF Telephoto 70–200 mm Canon zoom lens, and rendered using Zerene Stacker software v. 1.04. Images were enhanced using Photoshop 23.2.2.

## Morphological nomenclature

Differently from previous authors who used letters (*e.g.*, *Boulangé, 1924*; *Schulmeister, 2001*; *Schulmeister, 2003*) or numbers (*e.g.*, *Snodgrass, 1941*; *Alam, 1952*) to label the different muscle bundles, we follow *Daly (1964)*, *Friedrich & Beutel (2008)*, *Vilhelmsen (1996)*; *Vilhelmsen (2000a)*; *Vilhelmsen (2000b)*, and *Mikó et al. (2007)*; *Mikó et al. (2013)*, referring to the muscles as follows: the first component of the name refers to the site of origin, while the second refers to the site of insertion of the muscle. For example, the *proximoventral gonostyle/volsella complex-penisvalva muscle* is the muscle that is attached to the *gonostyle/volsella complex* and to the *penisvalva*. The proximoventral location differentiates it from the other *gonostyle/volsella complex-penisvalva* muscles.

For deciding the preferred term among synonyms, we follow the criteria established by *de Brito, Lanes & Azevedo (2021)*, with the integration of a new criterion (not in order of importance or priority): (1) the term that best represents the skeletal feature (shape and location of the body); (2) the term that is most widely used; (3) the term that was first introduced (oldest); (4) the term not employed also for other structures in other taxa (homonymy).

A list of the unified terminology employed here is provided in Tables 1–4 along with associated abbreviations and definitions included into an ontological framework.

## Morphological concepts

The insect cuticle is a continuous, acellular product of the single-layered outer epithelium (the *epidermis*) (*Hall, 1975*; *Adler, 1975*; *Denk-Lobnig & Martin, 2020*), consisting of comparatively rigid sclerites and comparatively flexible conjunctivae that alternate across the cuticle. The differences in flexibility of different cuticular regions allow the sclerites to change position relative to each other, enabling insects to achieve a wide range of motion types (*Mikó et al., 2013*; *Girón et al., 2023*).

The number, shape, and pattern of the sclerite-conjunctiva system varies between taxa and has changed throughout the course of evolution. One notable difference among taxa is sclerite "fusion" or "division"; the former occurs when two or more sclerites merge due to the disappearance of the separating conjunctiva, the latter by splitting of a pre-existing
**Table 1  Material examined with subfamily classification, locality information, repository, voucher number, and preservation method.**

| Subfamily | Taxon | Country | State/Province | Locality | Collector(s) | Repository | Voucher number | Preservation method |
|---|---|---|---|---|---|---|---|---|
| Cremastinae | *Temelucha* sp. | USA | Florida | Hal Scott Regional Preserve, Pine Flatwoods | D. Dal Pos & A. Pandolfi | UCFC | UCFMG_0000006 | Glycerol |
| Ichneumoninae | *Coelichneumon sassacus* (Viereck, 1917) | Canada | Manitoba | Whiteshell Prov. Pk., Pine Point Rapids Trail | Sharanowski lab | UCFC | UCFMG_0000001 | Point |
| Ichneumoninae | *Melanichneumon lissorufus* Heinrich, 1962 | Canada | Manitoba | Spruce Woods Prov. Pk. | Sharanowski lab | UCFC | UCFMG_0000002 | Point |
| Labeninae | *Labena grallator* (Say, 1835) | USA | Florida | Highland Co., Archbold Biological Station | Y. M. Zhang | UCFC | UCFMG_0000003 | Point |
| Labeninae | *Labena grallator* (Say, 1835) | USA | Florida | Hal Scott Regional Preserve, Cypress swamp | D. Dal Pos & A. Pandolfi | UCFC | UCFMG_0000016 | Glycerol |
| Mesochorinae | *Mesochorus* sp. | Canada | Manitoba | University of Manitoba, Points | UW/ELZ | UCFC | UCFMG_0000011 | Glycerol |
| Mesochorinae | *Mesochorus* sp. | Canada | Manitoba | University of Manitoba, Points | UW/ELZ | UCFC | UCFMG_0000014 | Glycerol |
| Mesochorinae | *Mesochorus* sp. | Canada | Manitoba | University of Manitoba, Points | UW/ELZ | UCFC | UCFMG_0000015 | Glycerol |
| Pimplinae | *Pimpla marginella* Brullé, 1846 | USA | Florida | Hal Scott Regional Preserve, Pine Flatwoods | D. Dal Pos & A. Pandolfi | UCFC | UCFMG_0000004 | Glycerol |
| Poemeninae | *Neoxorides pilosus* Townes, 1960 | Canada | Manitoba | Agassiz Prov. Pk. | Sharanowski lab | UCFC | UCFMG_0000012 | Glycerol |
| Tryphoninae | *Netelia* sp. | Canada | Manitoba | University of Manitoba, Points | UW/ELZ | UCFC | UCFMG_0000004 | Point |
| Tryphoninae | *Netelia* sp. | USA | Florida | Martin Co., Seabranch Preserve SP, Baygall | D. Serrano | UCFC | UCFMG_0000007 | Glycerol |
| Tryphoninae | *Netelia* sp. | USA | Arizona | Coconino Co., Tonto National Forest, 1 m W of Payson, 1440 m | D. Dal Pos & A. Pandolfi | UCFC | UCFMG_0000008 | Glycerol |
| Tryphoninae | *Netelia* sp. | USA | Arizona | Coconino Co., Tonto National Forest, 1 m W of Payson, 1440 m | D. Dal Pos & A. Pandolfi | UCFC | UCFMG_0000009 | Glycerol |
| Tryphoninae | *Netelia* sp. | Canada | Manitoba | Whiteshell Prov. Pk | Sharanowski lab | UCFC | UCFMG_0000017 | Glycerol |
| Rhyssinae | *Rhyssa persuasoria* (Linnaeus, 1758) | Canada | Manitoba | Whiteshell Prov. Pk. | Sharanowski lab | UCFC | UCFMG_0000010 | Glycerol |
| Xoridinae | *Xorides eastoni* (Rohwer, 1913) | Canada | Manitoba | Spruce Woods Prov. Pk | Sharanowski lab | UCFC | UCFMG_0000013 | Glycerol |

Dal Pos et al. (2023), *PeerJ*, DOI 10.7717/peerj.15874

Peer*J*

**Table 2  Anatomical terms used for skeletal features, cross-referenced to an ontological (formal) definition in the Hymenoptera Anatomy Ontology (HAO), linked through the HAO Uniform Resource Identifier (URI).**

| Abbreviation | Label | Definition | URI |
|---|---|---|---|
| S9 | Abdominal sclerite 9 | The abdominal sternum that is located on abdominal segment 9. | http://purl.obolibrary.org/obo/HAO_0000047 |
| aed | Aedeagus | The anatomical cluster that is composed of sclerites that are adjacent to the distal end of the ejaculatory duct | http://purl.obolibrary.org/obo/HAO_0000091 |
| ag | Apex gonostipitis | The apodeme that is located medially on the proximoventral margin of the gonostipes and is the site of origin of the ventral gonotyle/volsella complex-penisvalval muscles | http://purl.obolibrary.org/obo/HAO_0000134 |
| aps | Apiceps | The area that is the distal part of the gonossiculus and is connected to the parossiculus via membranous conjunctiva | http://purl.obolibrary.org/obo/HAO_0000141 |
| bsr | Basiura | The area that is the proximal part of the gonossiculus and corresponds to the site of insertion of medial penisvalvo-gonossicular muscle | http://purl.obolibrary.org/obo/HAO_0000179 |
| bs | Basivolsella | The area that is located on the parossiculus ventromedially of the cuspis | http://purl.obolibrary.org/obo/HAO_0001085 |
| c | Cupula | The sclerite that is connected via conjunctiva and attached via muscles to abdominal tergum 9 and the gonostyle/volsella complex | http://purl.obolibrary.org/obo/HAO_0000238 |
| cus | Cuspis | The projection that is located apicolaterally on the parossiculus and is adjacent to the digitus | http://purl.obolibrary.org/obo/HAO_0000239 |
| ejd | Ejaculatory duct | The duct that connects the vas deferens with the endophallus and is ectodermal in origin. | http://purl.obolibrary.org/obo/HAO_0000283 |
| end | Endophallus | The conjunctiva that connects the gonopore with the penisvalvae | http://purl.obolibrary.org/obo/HAO_0000291 |

Dal Pos et al. (2023), *PeerJ*, DOI 10.7717/peerj.15874

**Table 2** (*continued*)

| Abbreviation | Label | Definition | URI |
|---|---|---|---|
| erg | Ergot | The apodeme that is lateral, located medially on the penisvalva and corresponds to the sites of insertion of the lateral and distoventral gonostyle/volsella complex -penisvalval muscle and the parossiculo-penisvalval muscle | http://purl.obolibrary.org/obo/HAO_0000308 |
| fd | Fibula ducti | The sclerite that is located in the proximal end of the unpaired part of the ductus ejaculatorius | http://purl.obolibrary.org/obo/HAO_0000328 |
| fg | Foramen genitale | The anatomical space that is surrounded by the proximal margin of the cupula | http://purl.obolibrary.org/obo/HAO_0000346 |
| gnm | Gonomacula | The conjunctiva that is located at the distal apex of the harpe | http://purl.obolibrary.org/obo/HAO_0000382 |
| gss | Gonossiculus | The sclerite that is located on the distoventral part of the gonostyle/volsella complex, and is articulated with the more proximal sclerites of the gononstyle/volsella complex | http://purl.obolibrary.org/obo/HAO_0000385 |
| gst | Gonostipes | The sclerite that is located dorsolaterally on the gonostyle/volsella complex, is connected to the distal margin of the cupula, to the proximal margin of the harpe, and to the lateral margin of the volsella | http://purl.obolibrary.org/obo/HAO_0000386 |
| gsa | Gonostipital arm | The apodeme that is located proximally on the ventral part of the gonostipes | http://purl.obolibrary.org/obo/HAO_0000387 |
| gs | Gonostyle | The anatomical cluster that is composed of sclerites located distally of the cupula dorsolaterally of the volsella, and that surround the aedeagus | http://purl.obolibrary.org/obo/HAO_0000389 |

Dal Pos et al. (2023), *PeerJ*, DOI 10.7717/peerj.15874

**Table 2** (*continued*)

| Abbreviation | Label | Definition | URI |
|---|---|---|---|
| hrp | Harpe | The sclerite that is located distally on the gonostyle/volsella complex and does not connect to the cupula, and does not connect to the volsella by conjunctiva or muscles | http://purl.obolibrary.org/obo/HAO_0000395 |
| mss | Median sclerotized style | The sclerite that is located ventrally between the penisvalvae | http://purl.obolibrary.org/obo/HAO_0000531 |
| prp | Parapenis | The area that is the dorsomedian part of the gonostipes and is the site of origin of the distodorsal and proximodorsal gonostyle/volsella complex-penisvalval muscles | http://purl.obolibrary.org/obo/HAO_0000692 |
| pss | Parossiculus | The sclerite that is connected via conjunctiva distomedially to the gonostipes, and articulates with the gonossiculus | http://purl.obolibrary.org/obo/HAO_0000703 |
| ph | Phallotrema | The anatomical space that is the distal opening of the endophallus | http://purl.obolibrary.org/obo/HAO_0000714 |
| pv | Penisvalva | The sclerite that is in the middle of the external male genitalia, surrounds the distal part of the ductus ejaculatorius and the endophallus. | http://purl.obolibrary.org/obo/HAO_0000707 |
| pgp | Primary gonopore | The anatomical space that is the transition from the ductus ejaculatorius to the endophallus and therefore the transition from the internal to the external male genitalia. | http://purl.obolibrary.org/obo/HAO_0000821 |
| sp | Spatha | The sclerite that is unpaired and located just dorsally of the basal part of the aedeagus in some Aculeata. | http://purl.obolibrary.org/obo/HAO_0000942 |
| sv | Seminal vesicle | The anatomical space that functions as storage of spermatozoa | http://purl.obolibrary.org/obo/HAO_0001081 |

Dal Pos et al. (2023), *PeerJ*, DOI 10.7717/peerj.15874

**Table 2** (*continued*)

| Abbreviation | Label | Definition | URI |
| --- | --- | --- | --- |
| spc | Spiculum | The apophysis that is located medially on the anterior margin of the abdominal sternum 9 and corresponds to the site of origin of the mediolateral S9-cupulal muscles | http://purl.obolibrary.org/obo/HAO_0000946 |
| ts | Testis | The gonad that is consisting of testis follicles, is connected with the vas deferens and seminal vesicle | http://purl.obolibrary.org/obo/HAO_0001007 |
| vd | Vas deferens | The duct that connect the testis with the ejaculatory duct | http://purl.obolibrary.org/obo/HAO_0001052 |
| vvc | Valviceps | The area that is the distal part of the penisvalva dorsally of the ergot | http://purl.obolibrary.org/obo/HAO_0001047 |
| vvr | Valvura | The area that is located proximally of the ergot on the penisvalva. | http://purl.obolibrary.org/obo/HAO_0001050 |
| vol | Volsella | The anatomical cluster that is composed of the sclerites on the ventral part of the male genitalia that are not connected to the cupula via muscles | http://purl.obolibrary.org/obo/HAO_0001084 |

Dal Pos et al. (2023), *PeerJ*, DOI 10.7717/peerj.15874

**Table 3  Anatomical terms used for the muscles, cross-referenced to an ontological (formal) definition in the Hymenoptera Anatomy Ontology (HAO), linked through the HAO Uniform Resource Identifier (URI).**

| Abbreviation | Label | Definition | URI |
|---|---|---|---|
| c-gsdl | Dorsolateral cupulo-gonostyle/volsella complex | The cupulo-gonostyle/volsella complex muscle that arises from the dorsolateral part of the cupula, just laterally of the site of origin of the dorsomedian cupulo-gonostipal muscle, and inserts on the dorsolateral part of the gonostipes | http://purl.obolibrary.org/obo/HAO_0000278 |
| c-gsdm | Dorsomedial cupulo-gonostyle/volsella complex muscle | The cupulo-gonostyle/volsella complex muscle that inserts medially on the dorsal region of the gonostyle/volsella complex | http://purl.obolibrary.org/obo/HAO_0000279 |
| c-gsvl | Ventrolateral cupulo-gonostyle/volsella complex muscle | The cupulo-gonostyle/volsella complex muscle that inserts ventrolaterally on the gonostyle/volsella complex between the site of insertion of the ventromedial and dorsolateral cupulo-gonostyle/volsella complex muscles. | http://purl.obolibrary.org/obo/HAO_0001074 |
| c-gsvm | Ventromedial cupulo-gonostyle/volsella complex muscle | The cupulo-gonostyle/volsella complex muscle that inserts medially on the ventral region of the gonostyle/volsella complex. | http://purl.obolibrary.org/obo/HAO_0001075 |
| gn-pssd | Distal gonostipo-parossicular muscle | The gonostyle/parossicular muscle that arises distally of the lateral part of the gonostipes and inserts on the distal part of the parossiculus distally of the site of origin of the proximal gonostipo-parossicular muscle | http://purl.obolibrary.org/obo/HAO_0000247 |
| gn-pssp | Proximal gonostipo-parossicular muscle | The gonostyle-parossicular muscle that arises proximally from the lateral part of the gonostipes and inserts on the proximal part of the parossiculus | http://purl.obolibrary.org/obo/HAO_0000876 |
| gs-gs | Intragonostyle muscle | The muscle that connects the ventral and dorsal walls of the gonostyle basally | http://purl.obolibrary.org/obo/HAO_0002581 |
| gs-pss | Gonostyle/volsella complex-parossicular muscle | The muscle that arises ventromedially from the gonostyle, is proximomedially oriented, and inserts on the proximalmost sclerite of the volsella. | http://purl.obolibrary.org/obo/HAO_0002041 |
| gs-pvdd | Distodorsal gonostyle/volsella complex - penisvalval muscles | The dorsal gonostyle/volsella complex-penisvalval muscle that arises distodorsally from the gonostyle volsella complex and inserts on the proximal region of the penisvalva | http://purl.obolibrary.org/obo/HAO_0000250 |
| gs-pvdv | Distoventral gonostyle/volsella complex - penisvalval muscle | The ventral gonostyle/volsella complex-penisvalval muscle that arises from the proximoventral part of the gonostyle/volsella complex, inserts medially on the penisvalva and is oriented distodorsally | http://purl.obolibrary.org/obo/HAO_0000251 |

*(continued on next page)*

Dal Pos et al. (2023), *PeerJ*, DOI 10.7717/peerj.15874

**Table 3** (*continued*)

| Abbreviation | Label | Definition | URI |
|---|---|---|---|
| gs-pvl | Lateral gonostyle/volsella complex-penisvalval muscle | The gonostyle/volsella complex-penisvalval muscle that arises anterolaterally of the site of origin of the distodorsal gonostyle/volsella complex-penisvalval muscle and inserts laterally on the penisvalva | http://purl.obolibrary.org/obo/HAO_0000472 |
| gs-pvpd | Proximodorsal gonostyle/volsella complex - penisvalva muscle | The dorsal gonostyle/volsella complex penisvalval muscle that arises proximodorsally from the gonostyle/volsella complex and inserts on the penisvalva distally of the site of insertion of the distodorsal gononstyle/volsella complex-penisvalva muscle. | http://purl.obolibrary.org/obo/HAO_0000877 |
| gs-pvpv | Proximoventral gonostyle/volsella complex - penisvalval muscle | The ventral gonostyle/volsella complex-penisvalval muscle that arises ventromedially from the gonostyle/volsella complex, inserts on the proximal end of the penisvalva and is oriented proximodorsally | http://purl.obolibrary.org/obo/HAO_0000879 |
| gss-ph | Gonossiculo-phallotremal muscle | The muscle that arises from the gonossiculus and inserts on the phallotrema. | http://purl.obolibrary.org/obo/HAO_0002577 |
| ha-gon | Harpo-gonomaculal muscle | The male genitalia muscle that arises form the harpe and inserts on the gonomacula | http://purl.obolibrary.org/obo/HAO_0000396 |
| gs-hra | Apical gonostyle/volsella complex - harpal muscles | The gonostyle/volsellal complex-harpal muscle that arises from the distolateral margin of the gonostyle/volsellal complex and inserts on the lateral wall of the harpe | http://purl.obolibrary.org/obo/HAO_0000246 |
| ga-hrd | Distal gonostyle/volsella complex-harpal muscle | The gonostyle/volsella complex-harpal muscle that inserts on the median wall of the harpe and arises distally of the site of origin of the proximal gonostyle/volsella complex-harpal muscle | http://purl.obolibrary.org/obo/HAO_0000336 |
| gs-hrp | Proximal gonostyle/volsella complex-harpal muscle | The gonostyle/volsella complex-harpal muscle that inserts on the median wall of the harpe and arises proximally of the site of origin of the distal gonostyle/volsella complex-harpal muscle | http://purl.obolibrary.org/obo/HAO_0000926 |
| imvl | Median gonostyle/volsella complex-volsella muscle | The gonostyle/volsella complex-volsellal muscle that arises medially of the submedian conjunctiva on the distoventral margin of gonostyle/volsella complex | http://purl.obolibrary.org/obo/HAO_0000473 |
| imvll | Lateral gonostyle/volsella complex-volsella muscle | The gonostyle/volsella complex-volsellal muscle that arises laterally of the submedian conjunctiva on the distoventral margin of gonostyle/volsella complex | http://purl.obolibrary.org/obo/HAO_0002580 |

Dal Pos et al. (2023), *PeerJ*, DOI 10.7717/peerj.15874

**Table 3** (*continued*)

| Abbreviation | Label | Definition | URI |
|---|---|---|---|
| imvm | Gonostyle/volsella complex-gonossiculus muscle | The gonossicular muscle that arises ventromedially from the gonostyle/volsella complex and inserts laterally on the gonossiculus | http://purl.obolibrary.org/obo/HAO_0000517 |
| pss-ph | Parossiculo-phallotremal muscle | The male genitalia muscle that originates from the parossiculus and inserts on the endophallic membrane around the phallotrema | http://purl.obolibrary.org/obo/HAO_0000702 |
| pss-pv | Parossiculo-penisvalval muscle | The male genitalia muscle that arises from the proximal apex of the parossiculus and inserts medially on the penisvalva. The muscle inserts on the ergot if present. | http://purl.obolibrary.org/obo/HAO_0000701 |
| pv-gssl | Lateral penisvalvo-gonossicular muscle | The penisvalvo gonossicular muscle that is lateral to the medial penisvalvo-gonossicular muscle and attaches to the apiceps | http://purl.obolibrary.org/obo/HAO_0002579 |
| pv-gssm | Medial penisvalvo-gonossicular muscle | The penisvalvo-gonossicular muscle that is medial to the lateral penisvalvo-gonossicular muscle and attaches to the basiura. | http://purl.obolibrary.org/obo/HAO_0002578 |
| pv-mss | Penisvalvo-median sclerotized style muscle | The muscle that attaches to the median sclerotized style and to the valvura. | http://purl.obolibrary.org/obo/HAO_0002582 |
| pv-ph | Penisvalvo-phallotremal muscle | The male genitalia muscle that arises from the medial surface of the proximal part of the penisvalva and inserts on the endophallus just around the phallotrema. | http://purl.obolibrary.org/obo/HAO_0000710 |
| pv-pv | Interpenisvalval muscle | The male genitalia muscle that connects the valvurae. | http://purl.obolibrary.org/obo/HAO_0000433 |
| S9-cl | Lateral S9-cupulal muscle | The S9-cupulal muscle that arises sublaterally from S9 and inserts medioventrally on the cupula | http://purl.obolibrary.org/obo/HAO_0000464 |
| S9-cm | Medial S9-cupulal muscle | The male genitalia muscle that arises from the spiculum and inserts on the gonocondyle | http://purl.obolibrary.org/obo/HAO_0000516 |
| S9-cml | Mediolateral S9-cupulal muscle | The cupulal muscle that arises medially from abdominal sternum 9 and inserts ventrolaterally on the cupula | http://purl.obolibrary.org/obo/HAO_0000533 |
| vl-vl | Intervolsellal muscle | The male genitalia muscle that connects the proximal part of parossiculi. | http://purl.obolibrary.org/obo/HAO_0000441 |
**Table 4  Anatomical terms used for the muscles and cross-referenced with terms (including numbers and letters) used by other authors.** An en-dash symbol (−) identified that that specific muscle has not been treated by the author in the given work.

| Abbreviation | Label | Boulangé (1924) | Peck (1937a); Peck (1937b) | Snodgrass (1941) | Alam (1952) | Schulmeister (2001); Schulmeister (2003) | Mikó et al. (2013) |
|---|---|---|---|---|---|---|---|
| S9-cm | Medial S9-cupulal muscle | a | A | 1 | 2 | a | − |
| S9-cml | Mediolateral S9-cupulal muscle | b | B | 2 | 3 | b | Mediolateral S9-cupulal muscle |
| S9-cl | Lateral S9-cupulal muscle | c | C | 3 | 1 | c | Lateral S9-cupulal muscle |
| c-gsvm | Ventromedial cupulo-gonostyle/volsella complex muscle | d | D | 4 | 7 | d | Ventromedial cupulo-gonostyle/volsella complex muscle |
| c-gsvl | Ventrolateral cupulo-gonostyle/volsella complex muscle | e | E | 5 | 4, 5 | e | Ventrolateral cupulo-gonostyle/volsella complex muscle |
| c-gsdl | Dorsolateral cupulo-gonostyle/volsella complex | f | F | 7 | − | f | Dorsolateral cupulo-gonostyle/volsella complex |
| c-gsdm | Dorsomedial cupulo-gonostyle/volsella complex muscle | g | G | 6 | 6 | g | Dorsomedian cupulo-gonostyle/volsella complex muscle |
| gs-pvpv | Proximoventral gonostyle/volsella complex - penisvalval muscle | h | H | 8 | 14 | h | Proximoventral gonostyle/volsella complex - penisvalva muscle |
| gs-pvdv | Distoventral gonostyle/volsella complex - penisvalval muscle | i | I | 9 | 15 | i | Distoventral gonostyle/volsella complex - penisvalval muscle |
| gs-pvdd | Distodorsal gonostyle/volsella complex - penisvalval muscles | j | J | 10 | 13 | j | Distodorsal gonostyle/volsella complex - penisvalval muscles |
| gs-pvpd | Proximodorsal gonostyle/volsella complex - penisvalval muscle | k | K | 11 | 16 | k | Proximodorsal gonostyle/volsella complex - penisvalva muscle |
| gs-pvl | Lateral gonostyle/volsella complex-penisvalval muscle | l | L | 12 | 18 | l | Lateral gonostyle/volsella complex-penisvalval muscle |
| pv-gssl | Lateral penisvalvo-gonossicular muscle | m | M | 22 | 12 | m | − |
| pv-gssm | Medial penisvalvo-gonossicular muscle | | | | − | n | − |
| pv-ph | Penisvalvo-phallotremal muscle | n | N | 24 | 17 | nb | − |
| gss-ph | Gonossiculo-phallotremal muscle | | | | − | nd | − |
| pss-ph | Parossiculo-phallotremal muscle | | | | − | nl | − |
| gs-pss | Gonostyle/volsella complex-parossicular muscle | | | − | − | o | Gonostyle/volsella complex-parossicular muscle |
| gn-pssp | Proximal gonostipo-parossicular muscle | o | O | 18 | 8 | o' | − |
| gn-pssd | Distal gonostipo-parossicular muscle | | | 20 | − | o'' | − |
| imvll | Lateral gonostyle/volsella complex-volsella muscle | p | P | 19 | 10 | p | Lateral gonostyle/volsella complex-volsella muscle |
| imvl | Median gonostyle/volsella complex-volsella muscle | q | Q | | | | |
| | | r | R | 21 | 9 | qr | Medial gonostyle/volsella complex-volsella muscle |
| imvm | Gonostyle/volsella complex-gonossiculus muscle | s | S | 23 | 11 | s | Gonostyle/volsella complex-gonossiculus muscle |
| pss-pv | Parossiculo-penisvalval muscle | si | − | − | − | si | Parossiculo-penisvalval muscle |
| gs-hrd | Distal gonostyle/volsella complex-harpal muscle | | | 16 | − | t' | Distal gonostyle/volsella complex-harpal muscle |
| | | t | T | | | | |
| gs-hrp | Proximal gonostyle/volsella complex-harpal muscle | | | 15 | − | t'' | Proximal gonostyle/volsella complex-harpal muscle |

**Table 4** (*continued*)

| Abbreviation | Label | Boulangé (1924) | Peck (1937a); Peck (1937b) | Snodgrass (1941) | Alam (1952) | Schulmeister (2001); Schulmeister (2003) | Mikó et al. (2013) |
|---|---|---|---|---|---|---|---|
| ga-hra | Apical gonostyle/volsella complex - harpal muscles | u | U | – | – | u | Distal gonostipes/volsella complex-harpal muscle |
| ha-gon | Harpo-gonomaculal muscle | v | V | 17 | – | v | – |
| gs-gs | Intragonostyle muscle | – | – | | – | w | – |
| pv-pv | Interpenisvalval muscle | x | – | 13 | – | x | – |
| vl-vl | Intervolsellal muscle | – | – | | – | y | – |
| pv-mss | Penisvalvo-median sclerotized style muscle | z | – | 14 | – | z | – |

single sclerite by the development of a conjunctiva across it. For instance, the single sclerite connected to the *cupula* in the male genitalia in Ichneumonidae (=*gonostyle*) has been interpreted as the result of the fusion of the *harpe* and the *gonostipes*, forming one single continuous structure, named *gonoforceps* by different authors (*Peck, 1937a*).

The appearance or disappearance of conjunctivae is why the term "complex" (*e.g.*, *gonostyle/volsella complex*) was introduced by *Ronquist & Nordlander (1989)* and has been widely used for describing anatomical ontologies (*Mikó et al., 2013*; *Aibekova et al., 2022*). The term "anatomical complex" refers to a sclerite present in a particular taxon, which occupies a region that in other taxa is occupied by multiple sclerites. For instance, the *volsella* and the *gonostyle* are two different completely separate sclerites in the subfamily Labeninae (Ichneumonidae), but they are partially or entirely continuous in Mesochorinae.

## Description format

To help researchers navigate the different terminologies in standard taxonomic descriptions and future evolutionary studies, we provide a detailed morphological treatment of Hymenoptera male genitalia elements using the following structure:

(1) **Labels** –A list of synonymous labels employed by various authors is provided following the preferred term. The first author listed after the term is either the person who coined the term or the one who applied it for the first time in Hymenoptera, followed by authors who employed the term afterward. Newly proposed synonyms are marked with an asterisk (*).

(2) **Concept** –A general, homology-free diagnosis of the element and its components with special emphasis on their connectedness and structural properties, *i.e.*, epistemological recognition criteria.

(3) **Definition** –The Aristotelian definition of the element. Aristotelian definitions are used to build ontologies as they represent universal statements (see *Vogt, Bartolomaeus & Giribet (2022)* for more discussion).

(4) **Discussion of terminology** –The review of the usage of terms referring to the element within Hymenoptera with special emphasis on Ichneumonoidea.

(5) **Preferred term** –the label that has been selected as preferred, using the above criteria.

(6) **Morphological variation in Ichneumonoidea** –overview of the variation in the elements as observed from dissected specimens or as described in previous literature.

(7) **Comments** –general comments on the anatomical elements.

Each of the main elements (*abdominal sternum 9, cupula, gonostyle, volsella*, and *penisvalva*) of the male genitalia can be composed of a single sclerite or divided into multiple sclerites. Sclerites can be further divided into regions (also called areas) with more or less well-defined boundaries. For the best modeling of this complex system, we listed these structures nested within each other. This should help the reader navigate across the different terms.

Main elements of the male genitalia are identified by a Roman numeral (*e.g.*, GONOSTYLE=III); followed by another number referring to individual sclerites of the element (*e.g.*, GONOSTIPES=III.1), and finally, a Latin letter identifies regions of the

sclerite (*e.g.*, PARAPENIS=III.1.a). For example, the *parapenis* (III.1.a) is an area of the *gonostipes* (III.1) which is one of the sclerites that compose the *gonostyle* (III).

A note of caution: in some dry specimens, regions can appear more definable and can be possibly misidentified as separate sclerites, instead of being simply areas of certain sclerites. Thus, wet specimens are critical for understanding where sclerites start and end and thus ontological alignment of terms.

## RESULTS

### A review of Ichneumonoid male genitalia

With more than 48,000 described species, Ichneumonoidea is one of the largest superfamilies of Hymenoptera (*Branstetter et al., 2018*) and comprises roughly a third of all recognized species of Hymenoptera (*Sharanowski et al., 2021*). It is divided into two families, Braconidae (>21,000 spp.) and Ichneumonidae (>25,000 spp.) (*Quicke, 2015*; *Yu, van Achterberg & Horstmann, 2016*; *Klopfstein et al., 2019*; *Sharanowski et al., 2021*).

Many lineages of the superfamily show incredible external variation in the male genitalia. However, the genital morphology in Ichneumonoidea remains substantially undescribed for almost all of the 48,000 species, with the exception of the genus *Netelia* Gray (Ichneumonidae, Tryphoninae) (*e.g.*, *Townes, 1939*; *Konishi, 1991*; *Konishi, 1992*; *Konishi, 1996*; *Konishi, 2010*; *Bennett, 2015*; *Konishi, Chen & Pham, 2022*). Some authors have included descriptions of the genitalia in occasional single species descriptions (*e.g.*, *Loan, 1974*; *Walker, 1994*; *Watanabe & Matsumoto, 2010*; *Watanabe, Taniwaki & Kasparyan, 2015*; *Sobczak et al., 2019*; *Brajković et al., 2010*).

The first description of the male genitalia of Ichneumonidae was provided by *Bordas (1893)*, who analyzed the internal and external genital organs of five taxa, while *Peck (1937a)* and *Peck (1937b)* provided the first, and so far only, extensive study of Ichneumonidae male genitalia, with 96 taxa analyzed and comments on muscles, sclerite movements, and homology statements. In Braconidae, *Seurat (1898)* and *Seurat (1899)* was the first to mention the genitalia, while *Snodgrass (1941)* briefly analyzed and compared them with those of Ichneumonidae. Subsequently, *Alam (1952)* provided a more detailed study on the skeleto-musculature for one species of the subfamily Braconinae, while *Karlsson & Ronquist (2012)* provided the first modern description of the male external genitalia of two braconid species in the subfamily Opiinae.

Despite the fact that *Schulmeister (2003)* demonstrated that male genitalia characters can be informative for the higher-level classification of basal Hymenoptera and that other authors have successfully applied these structures in genus-level phylogenetic studies within sawflies (*Malagon-Aldana et al., 2021*) and Apocrita (*Andena et al., 2007*; *Owen et al., 2007*; *Mikó et al., 2013*), external male genitalia are rarely employed in phylogenetic studies on Ichneumonoidea. Slightly more research has been devoted to the genital organs of Braconidae, but most, if not all, of these studies show inconsistent terminologies. Some cite synonymous terms that are no longer valid (as discussed by *Schulmeister (2001)*; *Schulmeister (2003)*), and others associate a valid term with referring to a different sclerite.

The first suprageneric classification of Ichneumonidae using copulatory organs was proposed by *Peck (1937b)* (based on his previous work (*Peck, 1937a*)), who emphasized

the importance of the *abdominal sternum 9* (= subgenital plate). Likewise, *Pratt (1939)* discussed the genital characters of a subset of Ichneumonidae, providing the first (and only) key to the tribes based solely on male genitalia.

Various genera of Braconidae were analyzed by *Tobias (1967)*, who was the first to explore differences between subfamilies, while the subfamily Aphidiinae was extensively studied by *Tremblay (1979)*, *Tremblay (1981)* and *Tremblay (1983)*. *Quicke (1988)* surveyed Braconinae, concluding that the characters could be potentially useful for higher-level classification in the subfamily. Later, *Maetô (1996)* assessed the inter-generic variation of external male genitalia in Microgastrinae, while more recently *Brajković et al. (2010)* and *Žikić et al. (2011)* did the same in Agathidinae.

Over the years, other authors have included male genitalia in their phylogenies of Ichneumonoidea or one of its two families (*e.g.*, *Wahl & Gauld, 1998*) but the degree to which these characters have been employed is minimal. For instance, the recent morphological phylogenetic analyses of Ichneumonidae subfamilies by *Bennett et al. (2019)* included only four characters of the male terminalia, of which only two belong to the genital capsule. Reasons for the lack of use of male genitalic characters in Ichneumonoidea phylogenetics are unclear but possibly are due to: (1) the Ichneumonoidea classification system is mostly based on females and the association with males has been proven to be challenging; (2) males are rarely dissected, and genitalia characters have never been thoroughly assessed, nor they have been employed in taxonomic studies; and (3) rampant terminological inconsistencies, coupled with the overall complexity of the male genitalia, have discouraged researchers from exploring the male genitalia in Ichneumonoidea. This is why it becomes paramount to overcome at least one of these impediments, providing for the first time a complete assessment of the terminology.

## Towards a unified terminology

The male reproductive organs of Hymenoptera are composed of internal and external structures, in continuity with each other (*Schulmeister, 2001*). The inner reproductive system is composed of the *testis, vas deferens, seminal vesicle, accessory gland*, and *ductus ejaculatorius*. The external male genitalia consist of five elements: (1) *abdominal sternum 9*; (2) *cupula*; (3) *gonostyle*; (4) *volsella*; and (5) *penisvalva* (Figs. 1, 2). Note that all structures of the external male genitalia and their historical terms are described in depth below (see 'Methods and Results'). According to *Schulmeister (2001)* and *Schulmeister (2003)*, there are also two other sclerites: (6) the *median sclerotized style*, which is a thin sclerite that lies on the median axis of the ventral side of the external genitalia between the two *penisvalvae*, and (7) the *fibula ducti*, which is a sclerite present in the proximal end of the *ductus ejaculatorius*.

The elements of the male genitalia are interconnected through a network of muscles that move the sclerites individually or in conjunction (*Duncan, 1939*; *Snodgrass, 1941*; *Snodgrass, 1957*; *Michener, 1956*; *Schulmeister, 2001*; *Schulmeister, 2003*; *Mikó et al., 2013*). Four major muscle groups have been identified within Hymenoptera: (1) *abdominal sternum 9* to *cupula*, which are usually three muscles that control the movement of the entire genital capsule; (2) *cupula* to *gonostyle*, which are usually four distinct bundles that

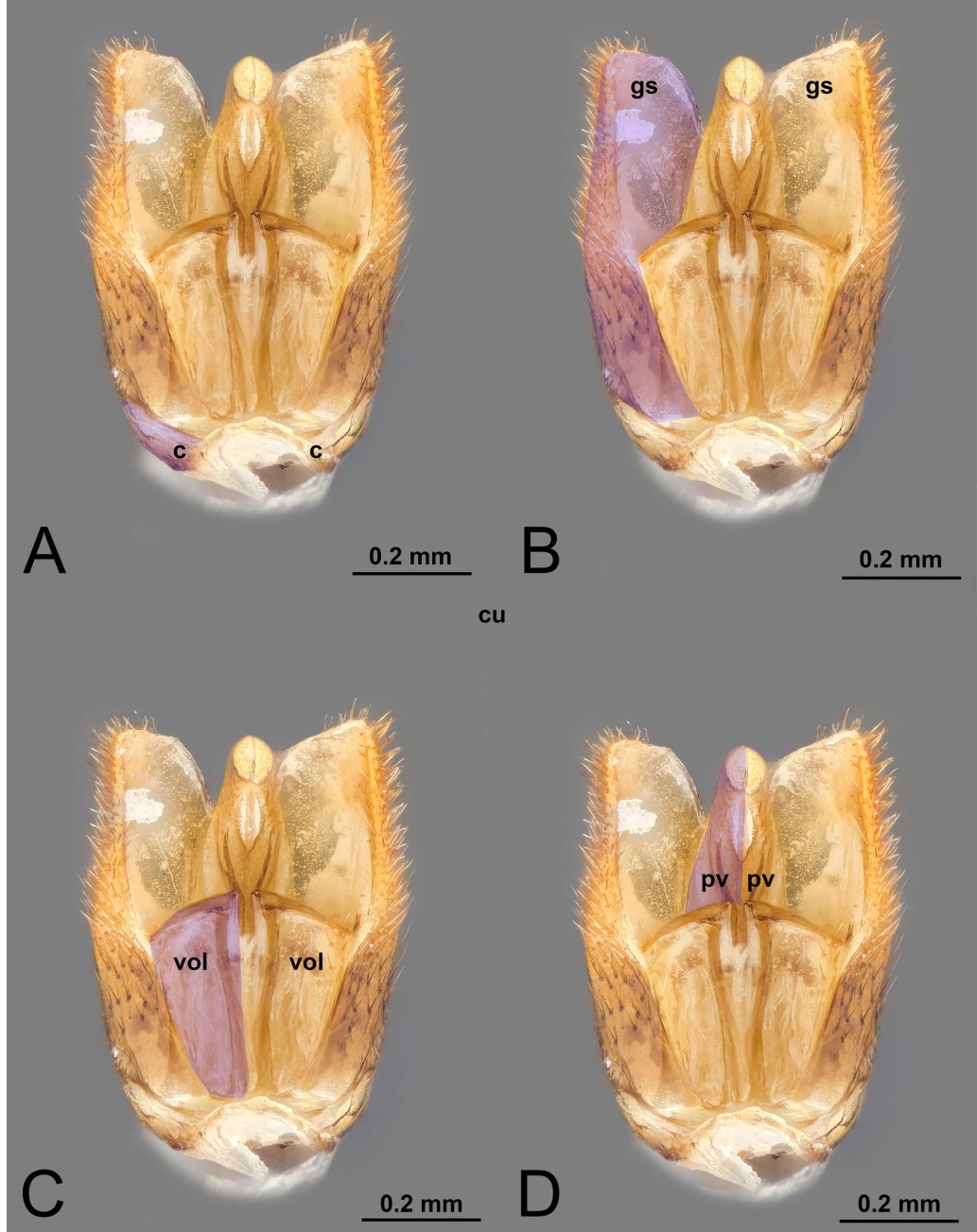

**Figure 1** Ventral view of male genitalia of *Melanichneumon lissorufus* (Ichneumonidae: Ichneumoninae) with different elements highlighted as follow: (A) Cupula (c). (B) Gonostyle (gs). (C) Volsella (vol). (D) Penisvalva (pv).

control the movement of the *gonostyles*; (3) *gonostyle* to *volsella*, which can be up to three distinct muscles that generally control the lateral motion of the *volsella* and the opening and closing of the apical clasping structure; (4) *gonostyle* to *penisvalva*, which usually consist of five distinct bundles, and control the motion of the *penisvalvae*.

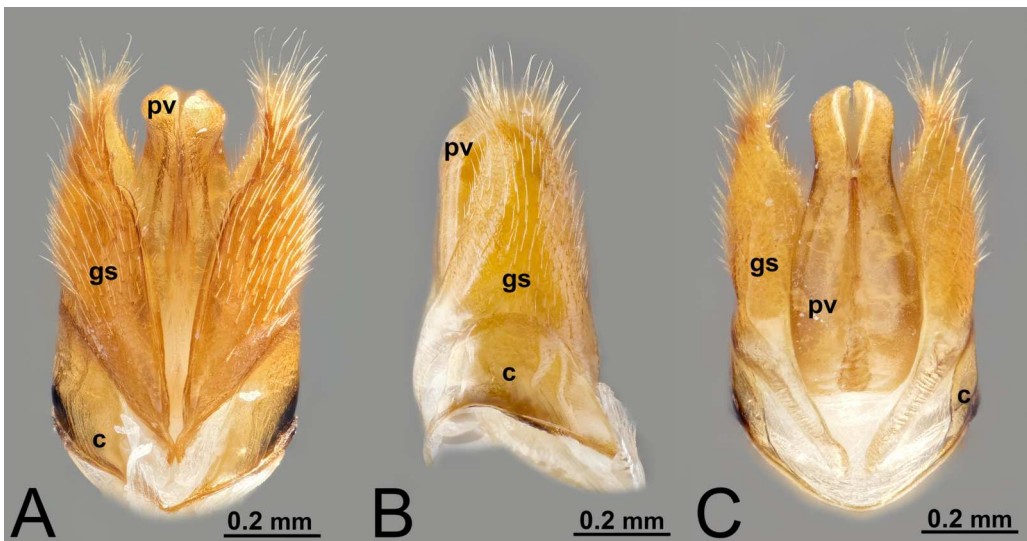

**Figure 2** Male genitalia of *Labena grallator* (Ichneumonidae: Labeninae). (A) Ventral view. (B) Lateral view. (C) Dorsal view.

## ABDOMINAL STERNUM 9

 **I. ABDOMINAL STERNUM 9**  (S9, Figs. 3A, 4A, 4C)
*ninth abdominal sternite* by *Worthley (1924)*; *Snodgrass (1935)*.
*subgenital plate*  by *Snodgrass (1935)*; *Watanabe & Matsumoto (2010)*.
*\*nono urotergite* by *Tremblay (1979)*; *Tremblay (1981)*; *Tremblay (1983)*.
*hypopygium* by *Pratt (1939)*; *Nichols (1989)*; *Karlsson & Ronquist (2012)*;
   *Broad, Shaw & Fitton (2018)*; *Bennett et al. (2019)*.
*hypopygidium* by *Nichols (1989)*; *Schulmeister (2001)*.
*hypandrium* by *Nichols (1989)*; *Peck (1937a)*; *Peck (1937b)*; *Karlsson & Ronquist (2012)*.
*annular lamina* by *Nichols (1989)*.
*hypotome* by *Nichols (1989)*.
*ninth sternal lobe* by *Nichols (1989)*.
*poculus* by *Nichols (1989)*.
*postgenital plate* by *Nichols (1989)*.
*metasomal sternum viii* by *Brothers & Carpenter (1993)*.
*abdominal sternum 9* by *Karlsson & Ronquist (2012)*.

*Concept.* The *abdominal sternum 9* is the ventral part of the ninth abdominal segment and connects the abdominal segments to the genital sclerites. Although not a direct component of the male genitalia, the *abdominal sternum 9* has a strong association with the *cupula*, by means of three major muscles: the *medial* (S9-cml), the *mediolateral* (S9-cm), and the *lateral S9-cupual* (S9-cl) *muscle* (Figs. 3A–3B, 4A–4B, 5A–5B; Tables 3–4). These muscles allow the protraction and retraction of the entire male genitalia. The *abdominal sternum 9* is usually produced proximo-medially into a process called the *spiculum* (spc,
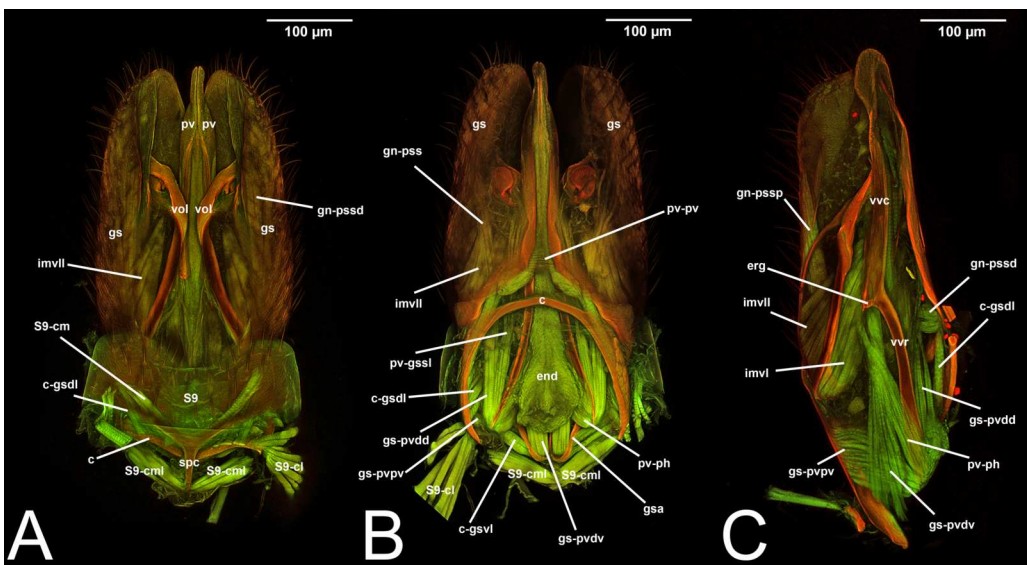

**Figure 3** CLSM volume rendered images of male genitalia of *Temelucha* sp. (Ichneumonidae: Cremastinae). (A) Ventral view. (B) Dorsal view. (C) Median view.

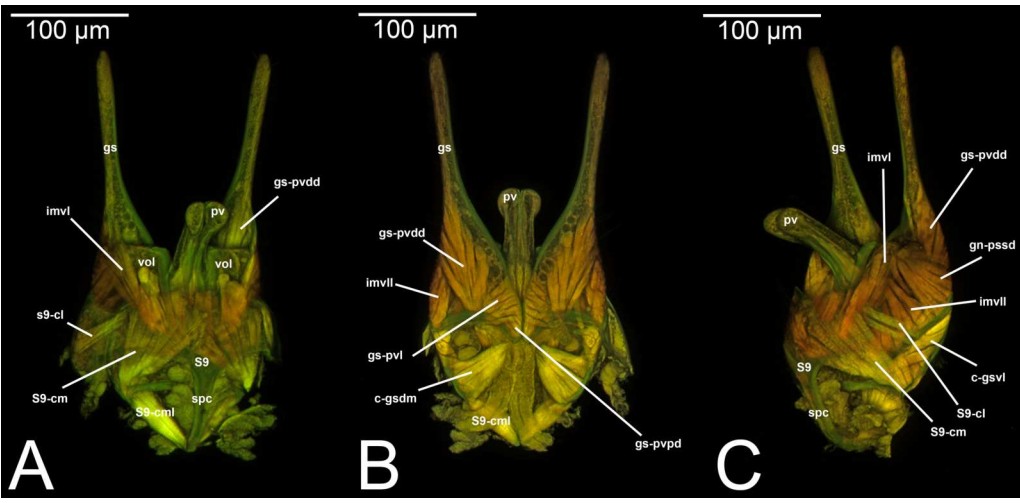

**Figure 4** CLSM volume rendered images of male genitalia of *Mesochorus* sp. (Ichneumonidae: Mesochorinae). (A) Ventral view. (B) Dorsal view. (C) Lateral view.

Figs. 3A, 4A, 4C), which is an apophysis that corresponds to the site of origin of the *mediolateral* (S9-cml) and *medial S9-cupular* (S9-cm) *muscles* (Figs. 3A–3B, Figs. 4A–4B).

   **Definition.** As defined by the HAO, the *abdominal sternum 9* is the abdominal sternum that is located on *abdominal segment 9* (Table 2).

   **Discussion of terminology.** Many of the terms employed are either variations of *abdominal sternum 9* (*e.g.*, *ninth sternal segment*) or have been rarely employed (*e.g.*, *poculus*). However, one of the most popular terms employed is *hypopygium*, which has

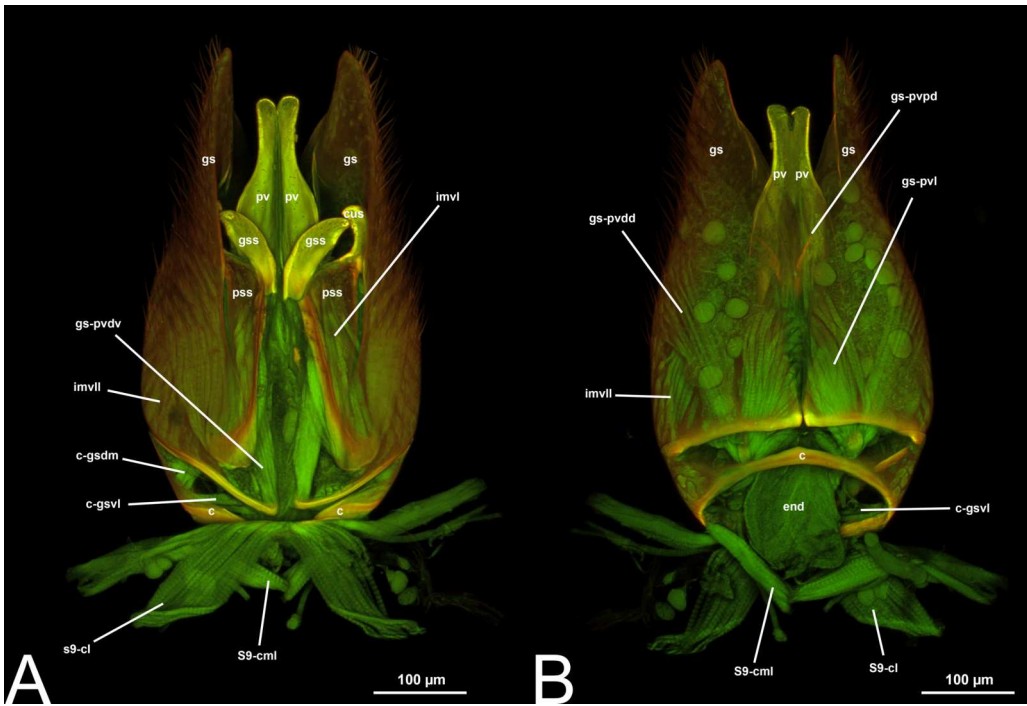

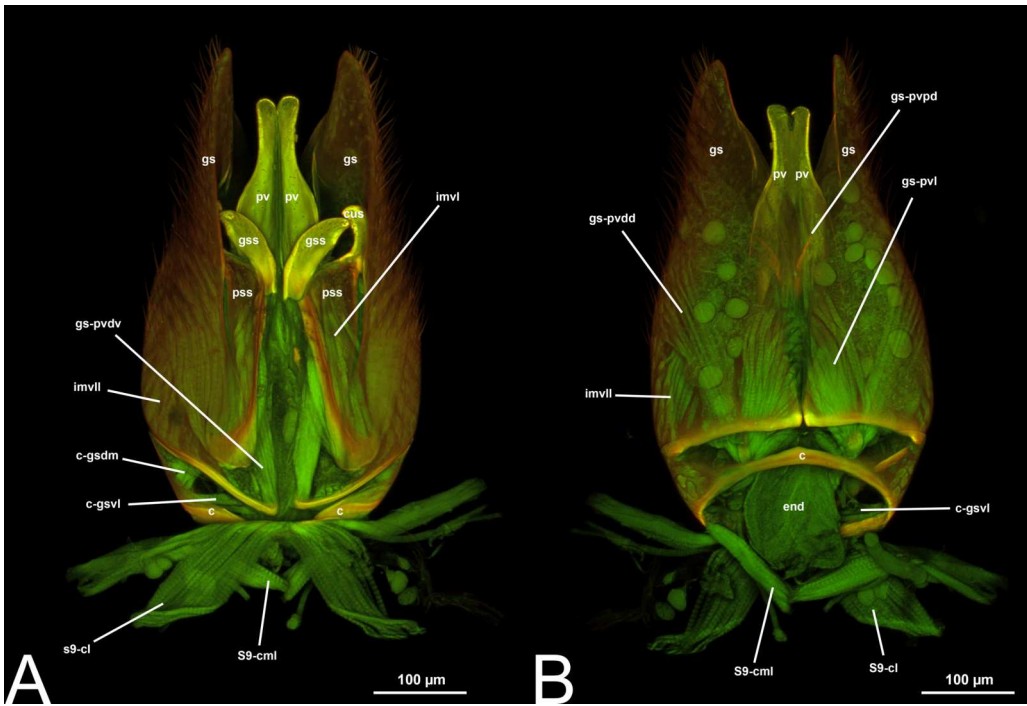

**Figure 5** CLSM volume rendered images of male genitalia of *Xorides eastoni* (Ichneumonidae: Xoridi-
nae). (A) Ventral view. (B) Dorsal view.

been employed to refer to the most posterior sternite in different insect orders. However,
from a morphological perspective, the *hypopygium* can be confusing. In fact, it has been used
within Hymenoptera to refer to the *abdominal sternum 7* in females (the last observable
sternite in females) or to the *abdominal sternum 9* in males (the last observable sternite in
males) (*e.g., Karlsson & Ronquist, 2012; Broad, Shaw & Fitton, 2018; Bennett et al., 2019*).
Moreover, *hypopygium* can also refer to different structures in different orders: in Diptera
it refers to the terminalia, in Lepidoptera to the multiple sclerites fused together, while
in Cicadomorpha (Hemiptera) it is used for the fused tergal and pleural parts of segment
9 (*Tuxen, 1956*). For these reasons, we strongly encourage using abdominal sternum 9,
which fulfills criterion 1 and 4.

Within Ichneumonoidea, *hypopygium* has been widely employed (*e.g., Pratt, 1939;
Karlsson & Ronquist, 2012; Broad, Shaw & Fitton, 2018; Bennett et al., 2019*), while
*hypandrium* was employed only by *Peck (1937a); Peck (1937b)* and by *Karlsson & Ronquist
(2012)*. Many other studies tend to exclude this sclerite as being part of the male genitalia
(*e.g., Quicke & van Achterberg, 1990*).

**Preferred term.** *Abdominal sternum 9.*

**Morphological variation in Ichneumonoidea.** According to several authors, the
*abdominal sternum 9* varies across the Ichneumonoidea. It has been used for differentiating
genera (*e.g., Heinrich, 1961*) and employed in phylogenetic reconstruction (*e.g., Bennett et
al., 2019*). There are several areas of variation: (1) the distal margin, which can be elongated,

flat, or concave (*Heinrich, 1961*; *Bennett et al., 2019*); (2) the shape of the *spiculum*, which can be extremely elongated (spc, Figs. 3A, 4A, 4C) or reduced, wide or thin (*Peck, 1937a*); (3) in the overall shape of the sclerite (see the image in *Peck, 1937a*, p. 246).

## II. CUPULA

**II. CUPULA**  (c, Figs. 1A, 2A–2C, 3A–3B, 5A–5B, 6B)
*cupule* by *Audouin (1821)*.
*hornringe* by *Hartig (1837)*.
*horny ring* by *Nichols (1989)*.
*pièce basilaire* by *Dufour (1841)*.
*kapsel* by *Schenck (1851)*.
*cardo* by *Thomson (1872)*.
*écailles* by *André (1879)*.
*lame basilaire* by *Bordas (1893)*.
*ringstück* by *Verhoeff (1832)*.
*plaque basilaire* by *Seurat (1898)*.
*sclerite accessorio* by *Berlese (1909)*.
*annular lamina* by *Wheeler (1910)*.
*pièce annulaire* by *Boulangé (1914)*.
*basalstück* by *Enslin (1918)*.
*gonocardo* by *Crampton (1919)*; *Peck (1937a)*; *Peck (1937b)*; *Pratt (1939)*.
*lamina annularis* by *Snodgrass (1941)*.
*gonobase* by *Michener (1944)*; *Michener (1956)*; *Delrio (1975)*; *Tremblay (1979)*; *Tremblay (1981)*; *Tremblay (1983)*; *Quicke & van Achterberg (1990)*; *Belokobylskij, Zaldivar-Riveron & Quicke (2004)*.
*basal ring* by *Crampton (1919)*; *Snodgrass (1935)*; *Snodgrass (1941)*; *Alam (1952)*; *Snodgrass (1957)*; *Tobias (1967)*; *Olmi (1984a)*; *Olmi (1984b)*; *Johnson (1984)*; *Quicke & van Achterberg (1990)*; *Konishi (1991)*; *Konishi (1992)*; *Konishi (1996)*; *Schulmeister (2001)*; *Schulmeister (2003)*; *Belokobylskij, Zaldivar-Riveron & Quicke (2004)*; *Konishi (2005)*; *Brajković et al. (2010)*.
*section 1* by *Smith (1969)*.
*\*guaina basale* by *Tremblay (1979)*; *Tremblay (1981)*; *Tremblay (1983)*.
*cupula* by *Birket-Smith (1981)*; *Schulmeister (2001)*; *Schulmeister (2003)*; *Karlsson & Ronquist (2012)*; *Mikó et al. (2013)*.

**Concept.** The *cupula* is an unpaired sclerite located at the proximal basis of the male genitalia. Different authors (see below in the Discussion of terminology) have consistently identified (with different names) an unpaired sclerite surrounding the proximal basis of the male genitalia. The *cupula* delimits a basal opening called the *foramen genitale* (*Snodgrass, 1941*; *Schulmeister, 2001*), and is connected via three muscles to the *abdominal sternum 9* and via four muscles to the *gonostyle* (Table 4).

**Definition.** As defined by the HAO, the *cupula* is the sclerite that is connected via conjunctiva and attached via muscle to the abdominal tergum 9 and the *gonostyle/volsella complex* (Table 2).

***Discussion of terminology.*** At least 21 terms have been introduced to refer to the *cupula*. Of these, only four (*basal ring, cupula, gonobase, gonocardo*) are worth discussing as they have been employed more than once after their introduction. All the other terms (see above), are rejected as they do not conform to criterion 2.

The first author to identify the basal sclerites of the genital organs in Hymenoptera was *Audouin (1821)*, who proposed the term *cupule* (later modified to *cupula* by *Birket-Smith (1981)*) to refer to an unpaired sclerite, which the author called the "*support commun* [= *general support*]".

In 1872, Thomson introduced *cardo* as a new term for the same structure after he studied the genus "*Bombis* [sic] [= *Bombus*]" and identified a basal capsule divided into two halves (*Thomson, 1872*). However, *Crampton (1919)* realized that *cardo* was also used for the basal sclerite of the maxillae in insects (rejected by criterion 4), and he proposed to replace *cardo* with *gonocardo* without realizing that another available term (*cupule*) has been already proposed. In the same work, *Crampton (1919)* introduced an additional term for the same sclerite, *basal ring*. This latter term, however, refers also to two different structures in two other orders: in Protura, it is used to refer to a basal structure of the male genitalia, while in Diptera, it is used for the combination between tergite and sternum IX (*Tuxen, 1970*). Since the term was first introduced in Diptera, and it is not equivalent to the one in Hymenoptera, *basal ring* should not be considered the preferred term (rejected by criterion 4).

Later on, *Michener (1944)*, following the periphallic theory, introduced *gonobase*, and subsequently, *Michener (1956)* formally synonymized *gonocardo* under *gonobase*, using the same rationale. Unfortunately, both authors failed to notice that the term *gonobase* corresponds to the non-equivalent *gonobasis* in basal insects (*e.g.*, *Willman, 1998*; *Schulmeister, 2001*), and, therefore, it should also not be used as the preferred term (rejected by criterion 4).

It must be noticed that there is a *cupula* also in Lepidoptera that was introduced by *Field (1950)* for a pair of pouches located on the 7th abdominal sternite of females (*Tuxen, 1956*). These two structures are not analogous to the *cupula* of Hymenoptera (HAO:0000238, see comments). Due to the date of introduction (the *cupula* of *Audouin (1821)* has been introduced before the *cupula* of *Field (1950)*) and due to the fact that the term has been specifically introduced in Hymenoptera, the *cupula* of *Audouin (1821)* can be retained and used as the preferred label for this sclerite (approved by criterion 3).

Within Ichneumonoidea, a number of authors used the term *gonobase* referring to the *cupula* (*Tremblay, 1979*; *Tremblay, 1981*; *Tremblay, 1983*; *Quicke & van Achterberg, 1990*; *Belokobylskij, Zaldivar-Riveron & Quicke, 2004*), and only two used *gonocardo* (*Peck, 1937a*; *Peck, 1937b*; *Pratt, 1939*) and *basal ring* (*Alam, 1952*; *Konishi, 2005*). As far as we know, the term *cupula* was applied to Ichneumonoidea only by *Schulmeister (2003)*.

***Preferred term.*** *Cupula.*

***Morphological variation in Ichneumonoidea.*** Across Ichneumonoidea, the *cupula* has undergone repeated reduction, sometimes within a subfamily. In *Labena grallator* (Say, 1845) (Ichneumonidae, Labeninae), *Xorides eastoni* (Rohwer, 1913) (Ichneumonidae, Xoridinae), and *Coelichneumon* Thomson, 1896 (Ichneumonidae, Ichneumoninae), the

*cupula* is well developed, especially on the lateral side (Figs. 2A–2B, 6A), while in *Mesochorus* Gravenhorst, 1829 (Ichneumonidae, Mesochorinae) and *Melanichneumon* Thomson, 1893 (Ichneumonidae, Ichneumoninae) (Fig. 1A) it is significantly reduced. Similar variations have also been observed by previous authors. *Peck (1937b)* described a ventrolaterally unusually broad *cupula* in *Megarhyssa macrura* Linnaeus, 1771 (Ichneumonidae, Rhyssinae) and *Banchus falcatorius* (Fabricius, 1775) (Hymenoptera, Banchinae), while *Pratt (1939)* noticed variations between members of the same subfamily, with *Delomerista* Förster, 1869, and *Perithous* Holmgren, 1859 (Ichneumonidae, Pimplinae) showing a well-developed and an extremely reduced *cupula*, respectively. The same pattern was observed in Braconidae, with some subfamilies (*e.g.*, Braconinae and Doryctinae) bearing a very elongated *cupula* (Figs. 77. 78 *in Quicke & van Achterberg, 1990*, p. 62), while in others (*e.g.*, Histomerinae) the same sclerite is reduced (Fig. 73 *in Quicke & van Achterberg, 1990*, p. 61).

***Comments on cupula.*** *Vilhelmsen (1997)* identified the *cupula* as a hymenopteran synapomorphy, but this result will likely require further investigation. In fact, other insect groups have a sclerite located basally to the rest of the sclerites and connected to the *abdominal sternum 9* (*e.g.*, the *phallobase* in Coleoptera, see *Snodgrass, 1957*, p. 30, fig. C).

 A final note concerns *Boudinot (2018)*, while he rejected the phallic theory, he did not introduce a new term for the *cupula*, but considered the structure a part of the fragmented base of genital appendages.

### III. GONOSTYLE

 **III. GONOSTYLE** (gs, Figs. 1–7)

*gonoforceps* (part.) by *Peck (1937a)*; *Peck (1937b)*; *Michener (1944)*; *Ross (1945)*;
 *Michener (1956)*; *Olmi (1984a)*; *Olmi (1984b)*; *Schulmeister (2001)*; *Schulmeister (2003)*;
 *Karlsson & Ronquist (2012)*; *Bennett et al. (2019)*.

\**squama* by *Townes (1939)*.

*paramere* (part.) by *Snodgrass (1957)*; *Pratt (1939)*; *Tobias (1967)*; *Olmi (1984a)*;
 *Olmi (1984b)*; *Quicke & van Achterberg (1990)*;
 *Belokobylskij, Zaldivar-Riveron & Quicke (2004)*; *Konishi (2005)*;
 *Watanabe & Matsumoto (2010)*; *Žikić et al. (2011)*; *Boudinot (2013)*;
 *Watanabe, Taniwaki & Kasparyan (2015)*; *Broad, Shaw & Fitton (2018)*;
 *Brajković et al. (2010)*.

*outer or central arm* by *Arnold (1951)*.

*gonopoden des 9 Segmentes* by *Haupt (1962)*.

*paramere exteriores* by *Priesner (1966)*.

\**claspers* by *Townes (1969a)*; *Townes (1969b)*; *Townes (1969c)*; *Dasch (1971)*; *Townes (1971)*;
 *Lee (1991)*.

*gonostyle* by *Bohart & Menke (1976)*; *Mikó et al. (2013)*.

*stipes* by *Birket-Smith (1981)*.

\**gonostipite* by *Tremblay (1983)*.

*latimere* by *Schulmeister (2001)*; *Schulmeister (2003)*.

\**gonopods* by *Boudinot (2018)*.
*Concept.*The *gonostyle* and the *volsella*, constitute the *gonostyle/volsella complex*. The *gonostyle* is the outermost structure of the male genitalia that is connected proximally with the *cupula* and medially with the *volsella*. When two clearly separated sclerites are present (the proximal *gonostipes* and the distal *harpe*), the entire structure has been named in different ways, mainly *paramere, gonostyle, latimere,* and *stipes* (see the list of synonymous terms above). However, when only one sclerite is present, only one term, *gonoforceps*, has been employed. Below we give an extensive explanation of why this has happened and why we propose *gonostyle* as the preferred term. Proximally, the *gonostyle* is elongated into two brace-like apodemes, called the *gonostipital arms* (gsa, Fig. 3B). At the tip of these, another apodeme, called *apex gonostipitis*, allows the insertion of two muscles, the *proximoventral* (gs-pvpv) and the *distoventral gonostyle/volsella complex-penisvalval muscle* (gs-pvdv) (Figs. 3B–3C, 5A; Tables 3–4). Historically a dorsomedial area has been identified, called *parapenis* (see below for an extensive treatment) that functions as a site of origin for the *proximodorsal* (gs-pvpd) and *distodorsal gonostyle/volsella complex-penisvalval muscle* (gs-pvdd) (Figs. 3B–3C, 4A–4C, 6B–6C, 7B; Tables 3–4).

*Definition.* As defined by the HAO, the *gonostyle* is the anatomical cluster that is composed of sclerites located distally of the *cupula*, dorsoventrally of the *volsella*, and that surrounds the *aedeagus* (Table 2).

*Discussion of terminology.* At least 12 terms have been introduced to refer to the *gonostyle*. Of these, five (*paramere, gonostyle, latimere, gonoforceps, claspers*) are worth discussing as they have been employed more than once after their introduction. The other terms (see above), are rejected as they do not conform to criterion 2.

Among the terms mentioned above, *paramere* has been the most widely used to refer to this cluster of sclerites of the male genitalia. The term has been applied across different insect orders (*e.g.*, Coleoptera), and within Hymenoptera is a renowned case of homonymy (*Yoder et al., 2010*). Applied for the first time in Hymenoptera by *Verhoeff (1832)* to refer to the *gonostipes+harpe+volsella*, the concept of *paramere* changed to refer only to the *penisvalva* (*Beck, 1933*; *Peck, 1937b*), then to the entire male genitalia (*cupula* excluded) (*Wheeler, 1910*), then only to the *harpe* (*Snodgrass, 1941*; *Königsmann, 1976*), and finally to *gonostipe+harpe* (*Snodgrass, 1957*). According to this already rampant confusion, *Peck (1937a)* coined a new term, *gonoforceps*, to refer to those specific cases in which the *harpe* and *gonostipes* are not distinguishable, and only a single continuous sclerite is present. *Gonoforceps* has been introduced specifically in Hymenoptera and has been very successfully applied in Ichneumonoidea. Later on, probably for the need to refer generally to a structure that could either be composed of two sclerites or just one, *Bohart & Menke (1976)* introduced the term *gonostyle* defining it as the "*outermost paired appendages of male genitalia, sometimes divided into basistyle and dististyle*" (see below for these last two terms).

*Schulmeister (2001)* first identified the confusion regarding homonyms of the term *paramere* and chose to reject its use. However, at the same time, she also introduced a new term, *latimere*, to identify the anatomical cluster formed by the *gonostipes+harpe*, maintained *gonoforceps* when the two sclerites were not distinguishable, and ignored the term *gonostyle*.

More recently, *Mikó et al. (2013)* employed the term *gonoforceps* in Ceraphronoidea when only one sclerite was discernible, but when the *harpe* and the *gonostipes* were present (as it is the case for the majority of the species), they employed the term *gonostyle*. On the other hand, *Boudinot (2013)* used *paramere*, instead of *gonoforceps*, even though a clear distally delimited sclerite (the *harpe*) is not present.

The term *claspers* were applied by *Townes (1969a)* to Ichneumonidae to refer to a special case in which the *gonostyle* is elongated, forming a rod (Figs. 4A–4C).

Among these many terms and concepts, it is not easy to identify which are preferred. However, some conclusions can be drawn. Following *Schulmeister (2001)*; *Schulmeister (2003)*, we reject the term *paramere* based on criterion 1 as it does not best represent the skeletal structure. At the same time, *claspers* should be rejected because, as pointed out by *Tuxen (1956)*, the term is widely used in many different other insect orders to refer to widely different structures, some of which are not homologous to the *gonostyle* (rejected by criterion 4). The term *latimere* also can be rejected as it has not been employed in any other work after *Schulmeister (2001)*; *Schulmeister (2003)* (rejected by criterion 2). Between *gonoforceps* and *gonostyle*, we suggest the latter as it better represents this skeletal area (fulfilling criterion 1). In fact, if the presence of the single sclerite has evolved through a fusion of *gonostipes* and *harpe*, it would be equivalent to the entire *gonostyle*, while if evolution resulted in a loss of the *harpe*, the *gonoforceps* would be equivalent to the *gonostipes* alone (*Schulmeister, 2001*). In both cases, the use of *gonoforceps* is unwarranted as there is no need for the employment of different terms to refer to a structure that is equivalent to either one or two sclerites. We certainly understand the "tradition" in using *gonoforceps*, but we also believe that to be able to move Hymenoptera and Ichneumonoidea into the phenomic era and allow future data mining for morphological features, it is better to avoid multiple terms for the same structure or group of sclerites, and employ accurate anatomical concepts (*Girón et al., 2023*). In this framework, the original definition of *gonostyle* proposed by *Bohart & Menke (1976)* allows the employment of the term in multiple cases: (1) when a clear delimitation between *harpe* and *gonostipes* is present; (2) when a clear delimitation between *harpe* and *gonostipes* is absent (the *gonoforceps*; see below for more details); and (3) when a delimitation between *harpe* and *gonostipes* is present, but it is not complete (see *Peck (1937a)*). Therefore, we strongly encourage the employment of the term *gonostyle* and use it as the preferred label.

As a side note, there is one term that was recently introduced by *Mikó et al. (2013)*: the *gonostyle/volsella complex*. It refers to the anatomical cluster composed of the sclerites located distally of the *cupula* and surrounding the *aedeagus*. We encourage the employment of this term especially when there is no clear delimitation between the *volsella* and the *gonostyles*.

*Peck (1937a)* and *Peck (1937b)* introduced the term *gonosquama* within the Ichneumonoidea. However, the term has not been used since then. On the other hand, the term *paramere* has been employed several times. Depending on the authors, *paramere* was used either to refer to the single sclerite that we now identify as a *gonostyle* (*Pratt, 1939*; *Tobias, 1967*; *Quicke & van Achterberg, 1990*; *Belokobylskij, Zaldivar-Riveron & Quicke, 2004*; *Žikić et al., 2011*; *Broad, Shaw & Fitton, 2018*; *Brajković et al., 2010*) or to indicate

just its distal part (*Alam, 1952*; *Tremblay, 1979*; *Tremblay, 1981*; *Tremblay, 1983*). The term *gonoforceps*, even though introduced purposely for Ichneumonidae, has only been used by three authors (*Peck, 1937a*; *Peck, 1937b*; *Karlsson & Ronquist, 2012*; *Bennett et al., 2019*). Finally, *claspers*, has been widely used in Ichneumonidae (*e.g.*, *Townes, 1969a*; *Townes, 1969b*; *Townes, 1969c*; *Townes, 1971*), especially when referring to the *gonostyle* of members in the subfamily Mesochorinae (*e.g.*, *Dasch, 1971*; *Lee, 1991*).

**Preferred term.** *Gonostyle.*

**Morphological variation in Ichneumonoidea.** Among the Ichneumonoidea herein surveyed and analyzed in the literature, the *gonostyle* show a small degree of variability. Within Ichneumonidae, the entire subfamily Mesochorinae (Figs. 4A–4C), and the genera *Lusius* Tosquinet, 1903 (Ichneumoninae) (DDP personal observation, 2023) and *Nematopodius* Granvenhorst, 1829 (Cryptinae) (G. Broad, personal communication, 2023) show a strong reduction of the apical part of the gonostyle, while *Pratt (1939)* found an apical knob in Rhyssinae (Ichneumonidae), hereby confirmed in *Rhyssa persuasoria* (Fig. 6C). Clear *gonostipital arms* are present in *Temelucha* (Fig. 3B), while they are inconsistent in *Mesochorus* (Fig. 4). In some Braconidae subfamilies (*e.g.*, Exothecinae), these sclerites likely underwent a shortening, exposing the *penisvalvae* and part of the *volsella* (Figs. 75, 76 *in Quicke & van Achterberg, 1990*, p. 61).

**III.1. GONOSTIPES** (Fig. 9A *in Mikó et al., 2013*, p. 275)
*stipes* by *Thomson (1872)*; *Zander (1900)*; *Wheeler (1910)*.
*gonostipes* by *Crampton (1919)*; *Peck (1937a)*; *Peck (1937b)*; *Ross (1937)*; *Ross (1945)*; *Schulmeister (2001)*; *Schulmeister (2003)*; *Mikó et al. (2013)*.
*pièce principale* by *Boulangé (1924)*.
*coxopodite* by *Beck (1933)*.
*basal part of forceps* by *Snodgrass (1941)*.
*basiparamere* by *Snodgrass (1941)*.
*lamina parameralis* by *Snodgrass (1941)*.
*\*parameral plate* by *Snodgrass (1941)*.
*gonocoxite* by *Michener (1944)*; *Michener (1956)*.
*basimere* by *Snodgrass (1941)*; *Boudinot (2013)*.
*basal part of stipes* by *Birket-Smith (1981)*.
*section 2* by *Smith (1969)*.
*basistyle* by *Bohart & Menke (1976)*.
*\*basiparamere* by *Brajković et al. (2010)*.
*\*gonocoxa* by *Boudinot (2013)*.

**Concept.** The *gonostipes* is the proximal sclerite composing the *gonostyle*. It has received several names, of which only one was applied to refer to two distinct concepts. When firstly introduced, the term *gonostipes* was applied to the *gonostipes*+*volsella* (*Crampton, 1919*), and only subsequently (*e.g.*, *Peck, 1937a*; *Peck, 1937b*; *Ross, 1937*; *Ross, 1945*; *Königsmann, 1976*) it was used to refer only to the *gonostipes*, excluding the *volsella*. The latter is the concept that has been more broadly and consistently employed (*Schulmeister, 2001*).

*Definition.* As defined by the HAO, the *gonostipes* is the sclerite that is located dorsolaterally on the *gonostyle/volsella complex*, and is connected to the distal margin of the *cupula*, to the proximal margin of the *harpe*, and to the lateral margin of the *volsella* (Table 2).

*Discussion of terminology.* At least 15 terms have been introduced to refer to the *gonostipes*. Of these, only three (*stipes, gonostipes, basimere*) are worth discussing as they have been employed more than once after their introduction. All the other terms (see above), are rejected as they do not conform to criterion 2.

The term *stipes* was introduced by *Thomson (1872)* to an area or sclerite of the male genitalia of *Bombus* Latreille (Hymenoptera: Apidae). As noted by *Schulmeister (2001)*, it is not clear if he referred to the *volsella* (which is extremely reduced in *Bombus*) or to the proximal sclerite of the *gonostyle*. *Crampton (1919)* replaced *Thomson*'s (*1872*) *stipes* with *gonostipes* acknowledging the fact that *stipes* is also employed to describe the appendages that bear the maxillary palps (rejected by criterion 4).

Later, *Michener (1944)*, following the periphallic theory, introduced the term *gonocoxite*, and, in 1957, Snodgrass employed the term *basimere* (firstly introduced by *Crampton (1942)* in Diptera) to refer to the *gonostipes* (*Snodgrass, 1957*), a term that was also preferred by *Boudinot (2013)*.

Among these terms, we strongly encourage the use of *gonostipes*, which is not only the oldest (fulfilling criterion 3) but also better represents the skeletal structure (fulfilling criterion 1). In fact, *gonocoxite* has never been employed after *Michener (1956)*, and, as already explained by *Schulmeister (2001)*, the term implies a homology with the coxa, and thus should be avoided (rejected by criteria 2 & 4). The term *basimere* is strongly connected to the concept of *paramere* (being its proximal sclerite), which, as we argued above, should be avoided (see under Discussion of terminology of the Gonostyle) (rejected by criteria 1). As already noticed by *Schulmeister (2001)*, *gonostipes* is a very well-known term and even though at its introduction, it was applied to the *gonostipes+volsella* complex, the modern concept of the term refers to the proximal sclerite of the *gonostyle*.

For the above reasons, the term *gonostipes* should be considered the preferred term.

The application of the term *gonostipes* is minimal in Ichneumonoidea due to the lack of a clear delimitation between the proximal sclerite and the *harpe* in most of the taxa. Only two authors used *gonostipes* (*Peck, 1937a*; *Peck, 1937b*; *Tremblay, 1979*; *Tremblay, 1981*; *Tremblay, 1983*).

*Preferred term. Gonostipes.*

*Morphological variation in Ichneumonoidea.* From our dissections, a clear delimitation of sclerites of the *gonostyle* is not evident in Ichneumonidae. However, *Peck (1937a)* acknowledged the presence of an articulated appendage in few ichneumonids. Further analyses, incorporating *Peck*'s (*1937a*) taxa, will be required to better understand the issue.

### III.1.a. PARAPENIS (prp, Figs. 6B, 7B)

*parapenis* by *Crampton (1919)*; *Boulangé (1924)*; *Schulmeister (2001)*; *Schulmeister (2003)*; *Boudinot (2013)*; *Koch & Liston (2017)*.
*manubrium* by *Crampton (1919)*.

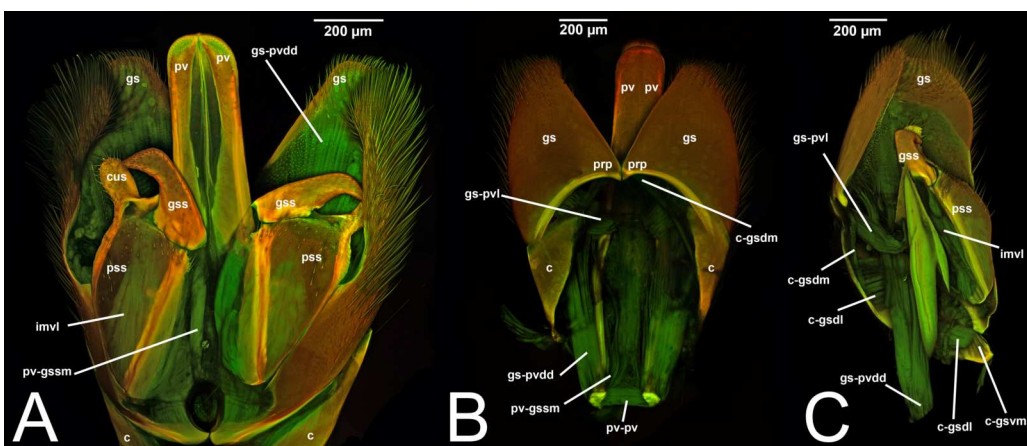

**Figure 6** CLSM volume rendered images of male genitalia of *Rhyssa persuasoria* (Ichneumonidae: Rhyssinae). (A) Ventral view. (B) Dorsal view. (C) Median view.

*praeputium by Crampton (1919).
*parapenis plate by Crampton (1919).
parapenial lobe by Evans (1950).
lobi parapenialis by Priesner (1966).

**Concept.** The *parapenis* is an area of the *gonostipes* located dorsomedially that has been mostly identified in basal Hymenoptera and very rarely in Apocrita (*e.g.*, *Schulmeister, 2003*). The area functions as a site of origin for *proximodorsal* (gs-pvpd) and *distodorsal gonostyle/volsella complex-penisvalval muscles* (gs-pvdd) (Tables 3–4). Only one concept (the original) has been applied to this area.

**Definition.** As defined by the HAO, the *parapenis* is the area that is the dorsomedial part of the *gonostipes* and is the site of origin of the *distodorsal* and *proximodorsal gonostyle/volsella complex-penisvalval muscles* (Table 2).

**Discussion of terminology.** *Crampton (1919)* introduced four terms to identify the area of *gonostipes*. Of these, only one, *parapenis*, has been employed more than once after its introduction by *Boulangé (1924)* and *Schulmeister (2001)*; *Schulmeister (2003)*. The other two terms, *parapenial lobe*, and *lobi parapenialis*, have not been employed further (rejected by criterion 2). Overall, the term *parapenis* has not been widely used, probably due to low variation across Apocrita, and to the lack of clear boundaries that delimit it. *Boudinot (2013)* decided not to use the term in Formicidae due to uncertain homology.

Within Ichneumonoidea, the term has not been employed, and the area has never been identified.

**Preferred term.** *Parapenis*.

**Morphological variation in Ichneumonoidea.** According to *Schulmeister (2003)*, the *parapenis* of Ichneumonidae is not set off from the rest of the *gonostipes*. In our observations, this is true for *Temelucha* (Fig. 3B), *Labena grallator* (Fig. 2A), *Mesochorus* (Fig. 4B), and

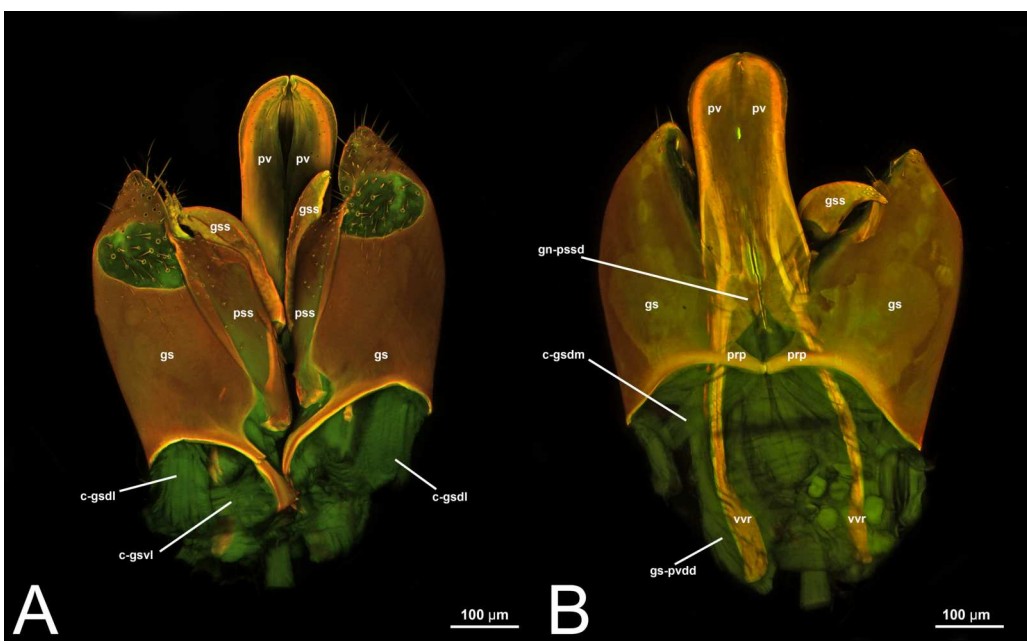

**Figure 7** CLSM volume rendered images of male genitalia of *Neoxorides pilosus* (Ichneumonidae: Poemeniinae). (A) Ventral view. (B) Dorsal view.

*Netelia* (Fig. 10B), but it is produced in the middle in Pomeninae (Fig. 7B) and Rhyssinae (Fig. 6B).

**III.2. HARPE**  (Fig. 9A *in Mikó et al., 2013*, p. 275)

*harpe* by *Crampton (1919)*; *Schulmeister (2001)*; *Schulmeister (2003)*; *Mikó et al. (2013)*.

*distal segment of gonopod* by *Crampton (1919)*.

*palette* by *Boulangé (1924)*.

*gonosquama* by *Peck (1937a)*; *Peck (1937b)*.

*paramere* (part.) by *Snodgrass (1941)*; *Michener (1944)*; *Alam (1952)*; *Michener (1956)*; *Königsmann (1976)*.

*gonostylus* by *Michener (1944)*; *Michener (1956)*.

*harpes* [sic] by *Ross (1945)*; *Wong (1963)*; *Königsmann (1976)*.

*telomere* by *Snodgrass (1957)*; *Boudinot (2013)*.

*harpago* by *Snodgrass (1957)*; *Yoshimura & Fisher (2011)*.

*squama* by *Townes (1957)*.

*disistyle* by *Bohart & Menke (1976)*.

*harpide* by *Audouin (1821)*; *Birket-Smith (1981)*.

**paramero* by *Tremblay (1979)*; *Tremblay (1981)*.

**lobo paramerale* by *Tremblay (1981)*; *Tremblay (1983)*.

*ventral paramere* by *Gobbi & Azevedo (2016)*.

**stylus* by *Boudinot (2018)*.

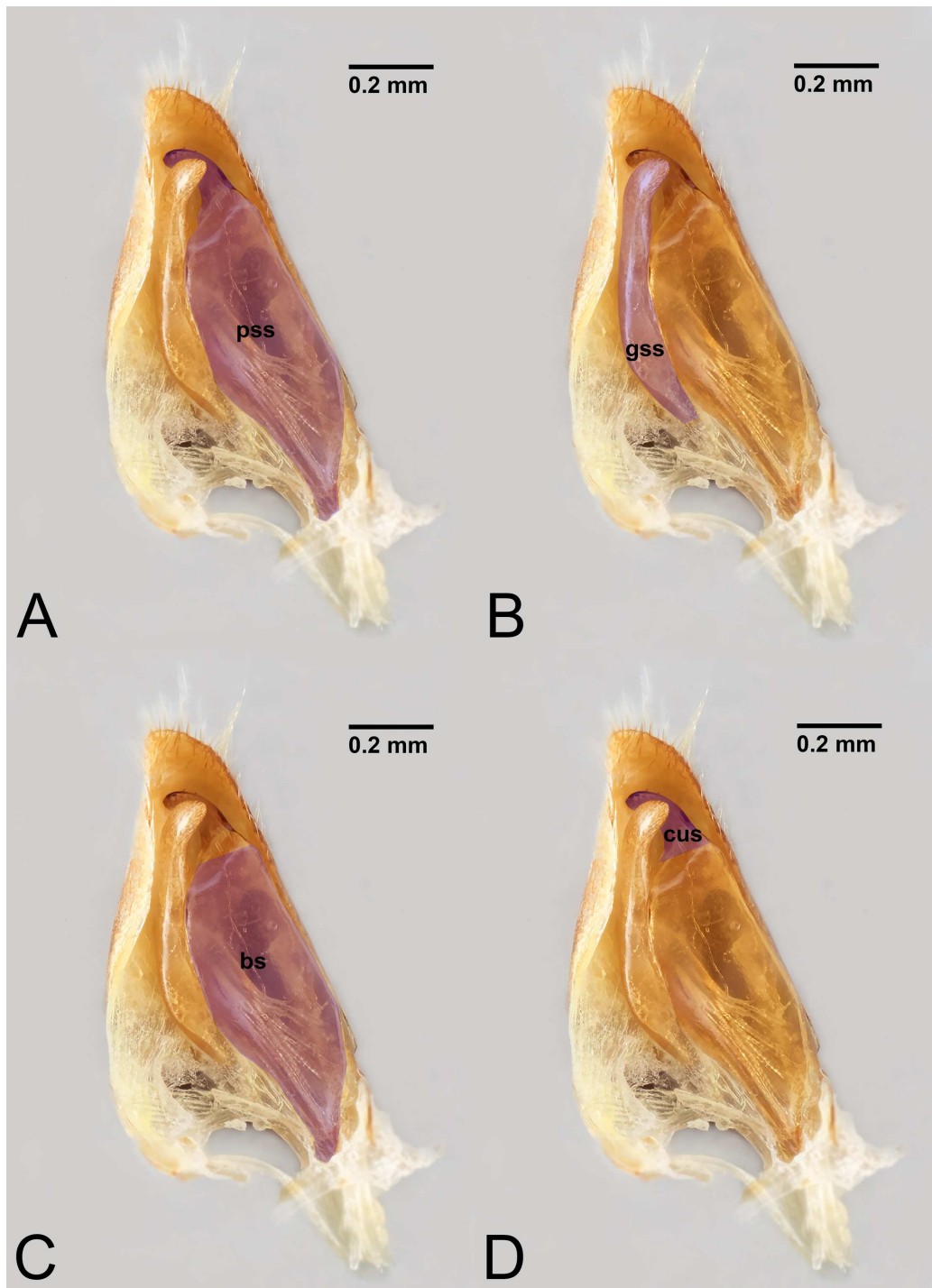

**Figure 8** Median view of volsella of *Labena grallator* (Ichneumonidae: Labeninae) with different elements highlighted as follow: (A) Parossiculus (pss). (B) Gonossiculus (gss). (C) Basivolsella (bas). (D) Cuspis (cus).

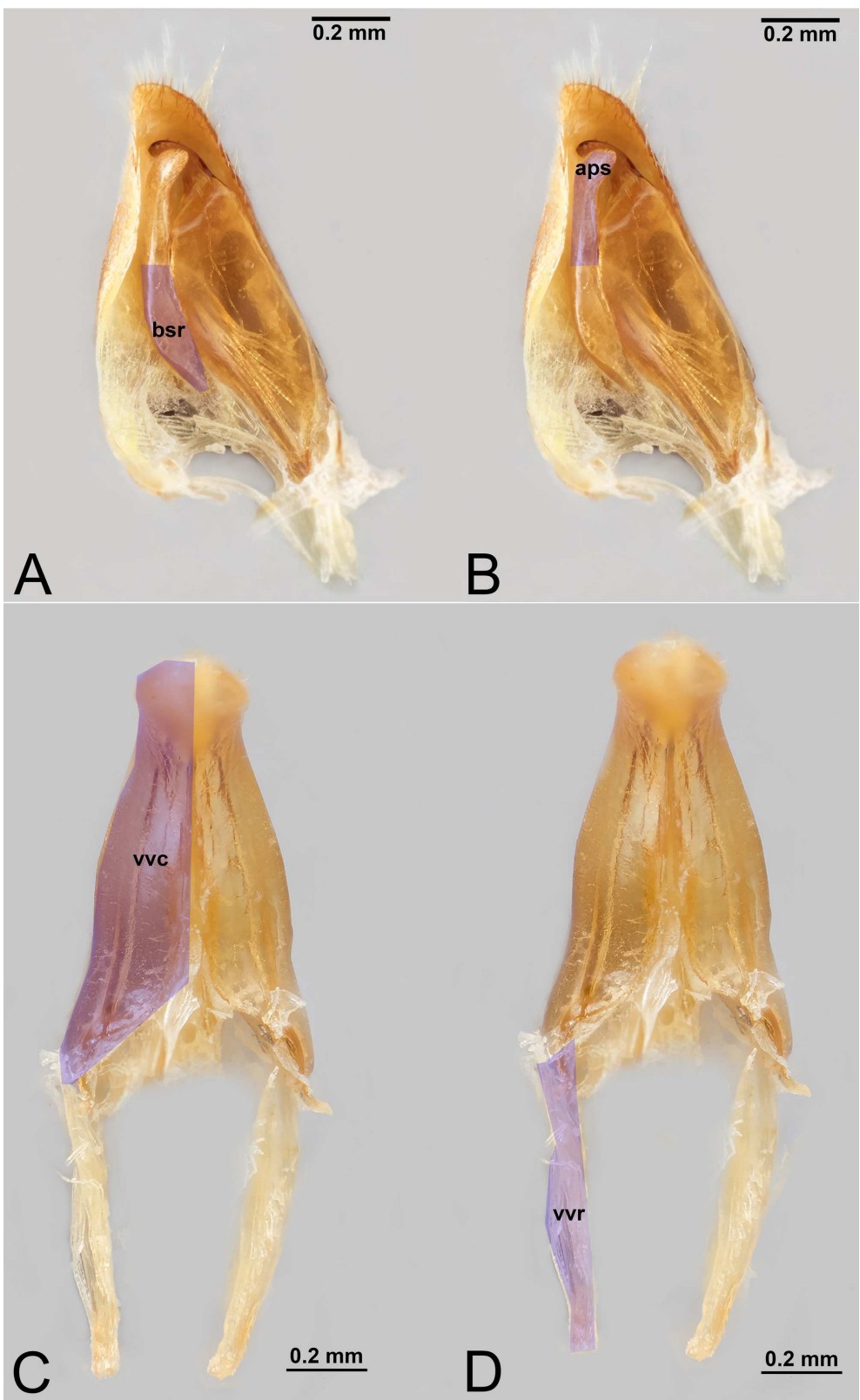

**Figure 9** Male genitalia of *Labena grallator* (Ichneumonidae: Labeninae) with different elements highlighted as follow: (A–B) Volsella, median view. (A) Basiura (bsr). (B) Apiceps (aps). (C–D) Penis-valva, ventral view. (C) Valviceps (vvc). (D) Valvura (vvr).

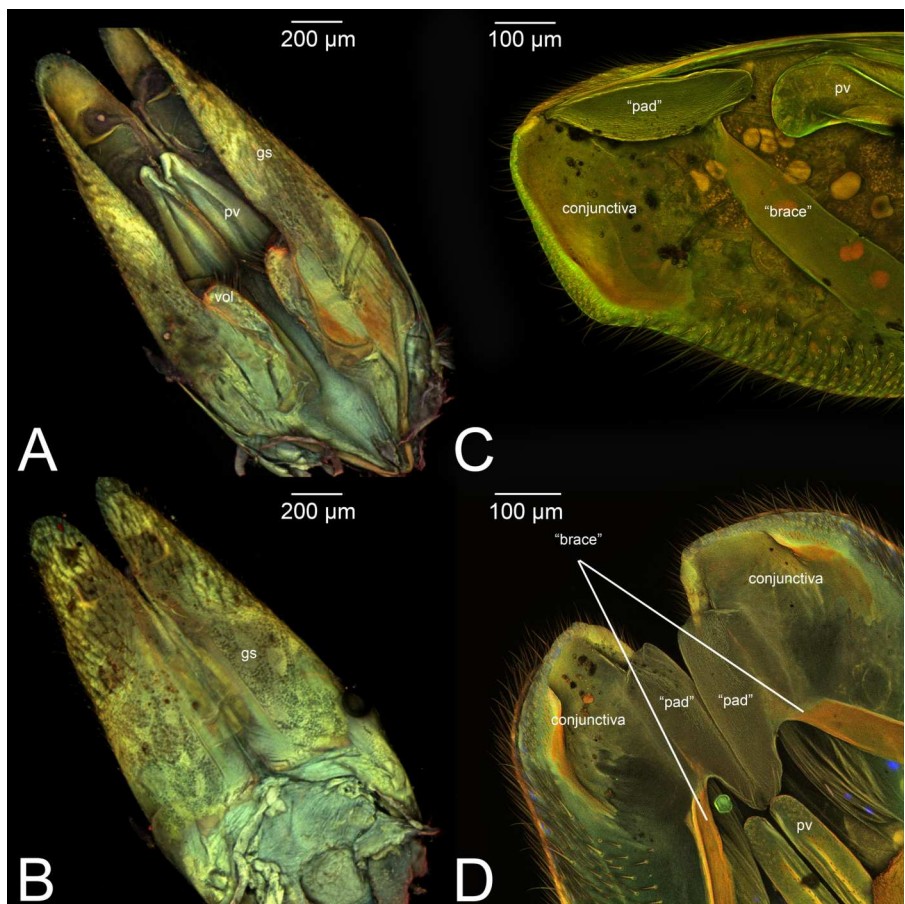

**Figure 10** CLSM volume rendered images of male genitalia of *Netelia* sp. (Ichneumonidae: Tryphoninae). (A) Ventral view. (B) Dorsal view. (C) Gonostyle, right apical view. (D) Gonostyle, apical view.

*Concept.* The *harpe* is the distally located sclerite of the *gonostyle* and is articulated via muscle with the *gonostipes*. On the *harpe* of Xyelidae, Pamphiliidae, Megalodontesidae, Siricidae, and Xiphydriidae, there is a conjunctiva called the *gonomacula* which seems to function as a suction cup (*Schulmeister, 2001*). Despite the many different terms applied to this sclerite, there has not been confusion regarding its identification and concept.

*Definition.* As defined by the HAO, the *harpe* is the sclerite that is located distally on the *gonostyle/volsella complex* and does not connect to the *cupula* nor to the *volsella* by conjunctiva or muscles (Table 2).

*Discussion of terminology.* At least 16 terms have been introduced to refer to the *harpe*. Of these, only five (*harpe, harpago, harpide, paramere, telomere*) are worth discussing as they have been employed more than once after their introduction. All the other terms (see above) are rejected as they do not conform to criterion 2.

The first to name the distal sclerite of the *gonostyle* was *Crampton (1919)*, who introduced the term *harpe*. Subsequently, *Peck (1937a)* acknowledged the presence of an articulated appendage in a few ichneumonids, which he called *gonosquama*, a term that has not been

used since then (rejected by criterion 2). *Snodgrass (1941)* decided to use *paramere* to refer only to the *harpe*, while *Ross (1945)*, followed by subsequent authors (*e.g.*, *Wong, 1963*; *Königsmann, 1976*) used the plural of *harpe* (*harpes*) to refer to the singular sclerite (and not to two sclerites). This is unwarranted and should be treated as a misspelling of *harpe*. In 1957, Snodgrass employed the term *telomere* (replacement for *distamire* introduced by *Crampton (1942)* in Diptera) to refer to the *harpe* (*Snodgrass, 1957*), a term that was also preferred by *Boudinot (2013)*. In the same work in which he introduced *telomere*, *Snodgrass (1957)* introduced another synonym, *harpago*, for the same structure. As far as we know, the latter term has been used only once after its introduction and can be rejected following criterion 2.

An interesting case is the term *harpide*, which was introduced by *Audouin (1821)* and then employed by *Birket-Smith (1981)* to replace *harpe*. However, the term, as explained by *Schulmeister (2001)*, was initially introduced by *Audouin (1821)* to identify a structure (not clear which one as there is no description nor images of it) of the male genitalia of bumblebees. Since bumblebees do not have a distally articulated sclerite, the *harpide* cannot refer to the *harpe*. Therefore, not only is *harpide* a synonym of *harpe* but it has been improperly applied. Of all these terms, we recommend the use of *harpe*, which is the oldest (fulfilling criterion 3) and that best represents the skeletal structure (fulfilling criterion 1). In fact, *paramere*, as we already discussed previously (see under Discussion of terminology of the *gonostyle*), suffers from extensive homonymy, and we strongly discourage its use (see also discussion in *Schulmeister (2001)*). At the same time, the term *telomere* is connected to the concept of *paramere* (being its distal sclerite), and therefore we also discourage its use.

As far as we know, the term *harpe* has never been used in Ichneumonoidea, and only *Peck (1937a)* employed the term *gonosquama* within Ichneumonidae (see below for more details).

**Preferred term.** *Harpe.*

**Morphological variation in Ichneumonoidea.** From our dissections, a clear delimitation of two individual sclerites is not evident in Ichneumonidae. However, *Peck (1937a)* acknowledged the presence of an articulated appendage in a few ichneumonids. Further analyses, incorporating *Peck*'s (*1937a*) taxa, will be required to better understand the issue.

***Comments on gonostyle and its associated elements.*** *Schulmeister (2001)* considered the absence of a *harpe* as a synapomorphy of the Vespina (Orussidae + Apocrita) and concluded that even when a distally delimited sclerite on the *gonostyle* is present in the Apocrita, it is probably not homologous with the *harpe* because there would be no associated musculature. However, *Mikó et al. (2013)* found a musculated *harpe* in some Ceraphronoidea and Trigonaloidea (both Apocrita) even though the arrangements of these muscles are different from that of the lower Hymenoptera. Their discovery was not enough to deduce homology of the *harpe* between Apocrita and lower Hymenoptera. *Boudinot (2013)* also found intrinsic muscles in approximately the same position in Formicidae.

It must be noted that *Boudinot (2018)* rejected the term *paramere sensu Snodgrass (1957)* because it was explicitly introduced within the phallic theory framework, and he instead employed the term *gonopods* for the same structures following the conclusion from the

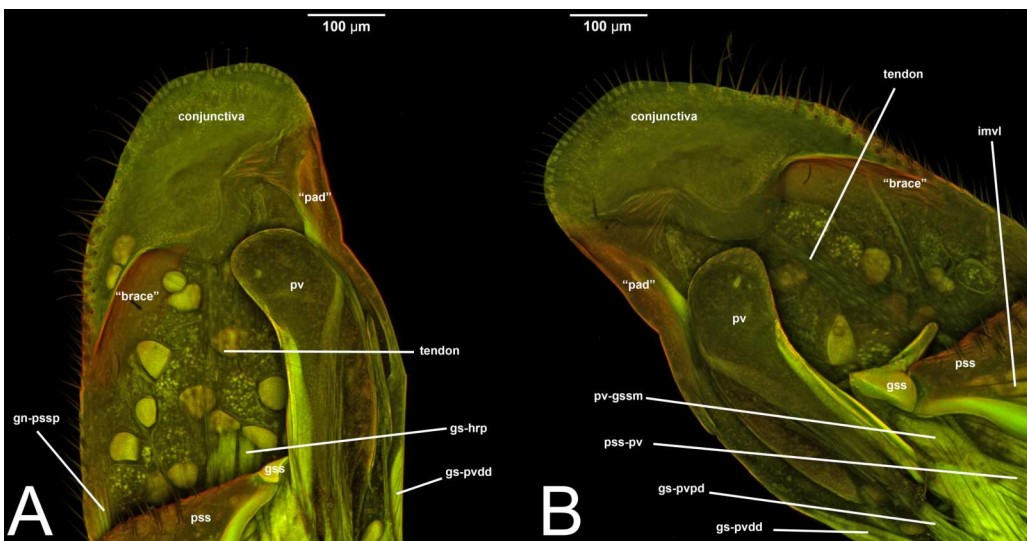

**Figure 11** CLSM volume rendered images of male genitalia of *Netelia* sp. (Ichneumonidae: Tryphoninae). (A) Right gonostyle, median view. (B) Left gonostyle, median view.

coxopod theory. Moreover, according to the same author, the *gonocoxa* (=*gonostipes* in the phallic theory), which is a fragment of the apical part of the sternum IX, fragmented a second time, forming the *parossiculus* (see below, under Volsella). According to this view, the *parossiculus* is just a secondary fragmentation of the *gonostipes* rather than part of the *volsella*. *Boudinot (2018)* also homologized the *harpe* with *stylus*, which he considered to have separated from the phallic apparatus (=*aedeagus*, see under Penisvalvae). This makes the *stylus* a sclerite with a completely different origin than the *gonostyle*. *Boudinot (2018)* did not provide any evidence or comments for the formation of a single sclerite.

## IV. VOLSELLA

**IV. VOLSELLA** (vol, Figs. 1C, 3A, 4A, 5A, 6A, 6C, 7A 8A–8B, 9A–9B, 10A, 11)

*volsella* by *Dufour (1841)*; *Peck (1937a)*; *Peck (1937b)*; *Pratt (1939)*; *Michener (1944)*;
  *Ross (1945)*; *Alam (1952)*; *Michener (1956)*; *Snodgrass (1941)*; *Snodgrass (1957)*;
  *Scobiola (1963)*; *Tobias (1967)*; *Königsmann (1976)*; *Johnson (1984)*; *Olmi (1984a)*;
  *Olmi (1984b)*; *Quicke & van Achterberg (1990)*; *Schulmeister (2001)*; *Schulmeister (2003)*;
  *Konishi (2005)*; *Žikić et al. (2011)*; *Karlsson & Ronquist (2012)*; *Boudinot (2013)*;
  *Mikó et al. (2013)*; *Broad, Shaw & Fitton (2018)*; *Brajković et al. (2010)*
*innere Haltzange* by *Enslin (1918)*.
*tenette* by *Snodgrass (1941)*.
*ossicle* by *Ross (1945)*.
*section 3* by *Smith (1969)*.
*section 3 of gonocoxite* by *Schulmeister (2001)*.

**Concept.** The *volsella*, together with the *gonostyle*, constitute the *gonostyle/volsella complex*, and it lies between the *gonostipes* and the *penisvalva*. Overall, the *volsella* consists

of two sclerites: (1) the *gonossiculus*, which is located distoventrally (gss, Fig. 8B); and (2) the *parossiculus*, located proximally (pss, Fig. 8A). The *parossiculus* is further divided into a basal and distal area: the *basivolsella* (bs, Fig. 8C) and the *cuspis* (cus, Fig. 8D), respectively.

The term *volsella* was introduced by *Dufour (1841)*, who did not clearly define which sclerite he was referring to. Over the years, it was employed for two main concepts, making it another case of homonymy (see the term *paramere*). For some authors (*e.g.*, *Crampton, 1919*; *Birket-Smith, 1981*), the *volsella* referred only to the sclerite connected with the *gonostipes*, and therefore it was identified with the *parossiculus*. For other authors (*e.g.*, *Snodgrass, 1941*; *Michener, 1944*; *Michener, 1956*), the *volsella* referred to the anatomical cluster composed of its proximal and its distal articulated sclerite; therefore, it included both the *parossiculus+gonossiculus*. This latter concept was employed more often after the redefinition by *Snodgrass (1941)*.

***Definition.*** As defined by the HAO, the *volsella* is the anatomical cluster composed of the sclerite on the ventral part of the male genitalia that is connected to the *cupula* via muscles (Table 2).

***Discussion of terminology.*** Few terms have been introduced to identify this group of sclerites, and all of them, except *volsella*, have never been used since their introduction. As *Schulmeister (2001)* pointed out, there is no suitable alternative to *volsella sensu Snodgrass (1941)* (see above). Therefore, given the priority (criterion 3) and the wide usage (criterion 2), *volsella* should be considered the preferred term.

Within Ichneumonoidea, many authors employed the term *volsella* to identify the entire anatomical cluster which lies between the *gonsotyles* and the *penisvalva* and does not articulate with the *cupula* (*Peck, 1937a*; *Peck, 1937b*; *Alam, 1952*; *Tobias, 1967*; *Konishi, 2005*; *Žikić et al., 2011*; *Karlsson & Ronquist, 2012*; *Brajković et al., 2010*). Only five identified it only with the *parossiculus* (*Pratt, 1939*; *Townes, 1939*; *Tremblay, 1979*; *Tremblay, 1981*; *Tremblay, 1983*; *Quicke & van Achterberg, 1990*; *Belokobylskij, Zaldivar-Riveron & Quicke, 2004*), while the two most recent studies that employed genitalia do not mention the term (*Broad, Shaw & Fitton, 2018*; *Bennett et al., 2019*).

***Preferred term.*** *Volsella.*

***Morphological variation in Ichneumonoidea.*** For the *volsella* we provided a brief account of the variability below each treatment of the sclerites or areas (see below), rather than a general observation of the cluster. This should guide future researchers to further analyses of the different parts.

### IV.1. PAROSSICULUS (pss, Figs. 5A, 6A, 6C, 7A, 8A, 11)

*parossiculus* by *Crampton (1919)*; *Schulmeister (2001)*; *Schulmeister (2003)*; *Mikó & Deans (2009)*; *Karlsson & Ronquist (2012)*; *Mikó et al. (2013)*.
*volsella* (part.) by *Wheeler (1910)*; *Crampton (1919)*; *Ross (1937)*; *Pratt (1939)*; *Tremblay (1979)*; *Birket-Smith (1981)*; *Tremblay (1981)*; *Tremblay (1983)*; *Quicke & van Achterberg (1990)*.
*pièce complémentaire* by *Boulangé (1924)*.
**cups* by *Tobias (1967)*.
*plaques volsellaires* by *Dessart & Gärdenfors (1985)*.

**Concept.** The *parossiculus* is the proximal sclerite of the *volsella* and is composed of two areas. (1) the *basivolsella* (bs, Fig. 8C), located basally; and (2) the *cuspis* (cus, Fig. 8D), which is a distally produced 'hook'.

**Definition.** As defined by the HAO, the *parossiculus* is the sclerite that is connected via conjunctiva distomedially to the *gonostipes* and articulates with the *gonossiculus* (Table 2).

**Discussion of terminology.** The term *parossiculus* was introduced by *Crampton (1919)* to refer to the proximal sclerite of the *volsella*. Similar to *volsella*, no better alternative to the term *parossiculus* has been proposed and employed. Therefore, given the priority (criterion 3) and the wide usage (criterion 2), *parossiculus* should be considered the preferred term.

Within Ichneumonoidea, the term *parossiculus* has been either wrongly identified with the entire *volsella* (*Pratt, 1939*; *Townes, 1939*; *Quicke & van Achterberg, 1990*; *Belokobylskij, Zaldivar-Riveron & Quicke, 2004*) or simply not recognized (*e.g.*, *Peck, 1937a*; *Peck, 1937b*). To our knowledge, only *Karlsson & Ronquist (2012)* employed *parossiculus* correctly to define the basal part of the *volsella*.

**Preferred term.** *Parossiculus*.

**Morphological variation in Ichneumonoidea.** According to *Peck (1937b)*, the *parossiculus* and the *gonostyle* are indistinguishable in some Cryptinae, Ichneumoninae, Ophioninae, and Tryphoninae, forming, therefore, a clear *gonostyle/volsella complex*. This is true also for the Mesochorinae (Fig. 4), but not for Labeninae (Figs. 8, 9A–9B), and Rhyssinae (Fig. 6A) in which the two are clearly distinguishable. The majority variability relies more on the presence/absence of the *cuspis*, which seems to vary according to the different subfamilies (see under Cuspis).

### IV.1.a. BASIVOLSELLA (bas, Fig. 8C)

*basivolsella* by *Peck (1937a)*; *Peck (1937b)*; *Ross (1945)*; *Alam (1952)*; *Olmi (1984a)*; *Olmi (1984b)*; *Schulmeister (2001)*; *Schulmeister (2003)*.
*lamina volsellaris* by *Snodgrass (1941)*; *Johnson (1984)*; *Maetô (1996)*; *Konishi (2005)*.
*\*lamina volsellare* by *Tremblay (1979)*; *Tremblay (1981)*; *Tremblay (1983)*.
*volsellar plate* by *Snodgrass (1941)*; *Snodgrass (1957)*.

**Concept.** The *basivolsella* is the proximal area of the *parossiculus*. There has not been much confusion regarding the identification of this area, even though its fiat boundaries led many authors (*e.g.*, *Karlsson & Ronquist, 2012*) to not discuss it in their work.

**Definition.** As defined by the HAO, the *basivolsella* is the area that is located on the *parossiculus* ventromedially to the *cuspis* (Table 2).

**Discussion of terminology.** There have been few issues regarding what term to apply to the proximal area of the *parossiculus*. In fact, *basivolsella* was the first term introduced (*Peck, 1937a*) and, by far, has been the most employed by different authors. Subsequently, *Snodgrass (1941)* referred to this area with the general term of *volsellar plate* and *lamina volsellaris*, which was not commonly employed afterwards. Therefore, given the priority (criterion 3) and the wide usage (criterion 2), *basivolsella* should be considered the preferred term.

Within Ichneumonoidea, the term *basivolsella*, despite being introduced specifically for the superfamily by *Peck (1937a)*, was subsequently applied only by *Peck (1937b)* and *Alam (1952)*. Other authors either did not recognize the structure (*e.g.*, *Karlsson & Ronquist, 2012*) or used the synonym *lamina volsellaris* (*Tremblay, 1979*; *Tremblay, 1981*; *Tremblay, 1983*; *Maetô, 1996*; *Konishi, 2005*).

**Preferred term.** *Basivolsella.*

**Morphological variation in Ichneumonoidea.** According to *Peck (1937b)*, the *basivolsella* can be used for delimiting subfamilies and various genera, and its dimension can be employed to easily separate several species groups of Ichneumoninae. From our observations, it is clear that the *basivolsellae* in *Labena grallator* and *Rhyssa persuasoria* are notably expanded (Figs. 6A, 8C), while it seems less so in *Temelucha* sp. (Fig. 3A). Further studies are needed to assess the accuracy of *Peck (1937b)*.

 **IV.1.b. CUSPIS**  (cus, Figs. 6A, 8D)

*distivolsella* by *Peck (1937a)*; *Peck (1937b)*; *Pratt (1939)*; *Olmi (1984a)*; *Olmi (1984b)*;
  *Schulmeister (2001)*; *Schulmeister (2003)*.
\*apical lobe of volsella by *Townes (1939)*.
*cuspis* by *Snodgrass (1941)*; *Michener (1944)*; *Alam (1952)*; *Michener (1956)*;
  *Snodgrass (1957)*; *Delrio (1975)*; *Tremblay (1979)*; *Birket-Smith (1981)*;
  *Tremblay (1981)*; *Tremblay (1983)*; *Maetô (1996)*; *Schulmeister (2001)*;
  *Schulmeister (2003)*; *Belokobylskij, Zaldivar-Riveron & Quicke (2004)*; *Konishi (2005)*;
  *Žikić et al. (2011)*; *Brajković et al. (2010)*.
*cuspis volsellaris* by *Snodgrass (1941)*.
\*cuspidal processes by *Tobias (1967)*; *Quicke & van Achterberg (1990)*.
\*cuspide by *Tremblay (1979)*; *Tremblay (1981)*; *Tremblay (1983)*.
\*cuspides by *Tremblay (1979)*; *Tremblay (1981)*; *Tremblay (1983)*.
\*lobo volsellare by *Tremblay (1979)*; *Tremblay (1981)*; *Tremblay (1983)*.

**Concept.** The *cuspis* is the distally located, elongated area of the *parossiculus*. There has been little confusion regarding the identification of this area, even though its arbitrary boundaries led many authors (*e.g.*, *Karlsson & Ronquist, 2012*) to not discuss it in their work.

**Definition.** As defined by the HAO, the *cuspis* is the projection that is located apicolaterally on the *parossiculus* and is adjacent to the *gonossiculus* (Table 2).

**Discussion of terminology.** There are two competing terms that could be applied to the distal area of the *parossiculus*, namely the *cuspis* and the *distivolsella*. The latter was introduced by *Peck (1937a)*, and it was employed in four subsequent papers (*Peck, 1937b*; *Pratt, 1939*; *Schulmeister, 2001*; *Schulmeister, 2003*). On the other hand, *cuspis* was introduced later on by *Snodgrass (1941)*, and it became more widely used (at least 16 papers).

*Schulmeister (2001)* noticed that there is no objective reason to prefer either *cuspis* or *distivolsella*. However, she discouraged the use of *distivolsella*, stating that when *basivolsella* and *distivolsella* are used together, they could imply that the two alone (without the

*gonossiculus*) actually compose the entire *volsella*. This misconception could lead to two consequences: (1) identifying the *volsella* only with the *parossiculus* (without the *gonossiculus*); and (2) identifying the *gonossiculus* as part of the *cuspis*. Therefore, she proposed the employment of *cuspis* to avoid such conceptual misunderstanding, a decision that has also been followed by *Mikó et al. (2013)* and *Boudinot (2013)*.

The reasoning adopted by *Schulmeister (2001)* seems to fulfill criterion 1 and 2. Therefore, *cuspis* should be considered the preferred term.

Within Ichneumonoidea, the term *cuspis* has been used regularly (*Alam, 1952*; *Tremblay, 1979*; *Tremblay, 1981*; *Tremblay, 1983*; *Maetô, 1996*; *Belokobylskij, Zaldivar-Riveron & Quicke, 2004*; *Konishi, 2005*; *Žikić et al., 2011*; *Brajković et al., 2010*), while *distivolsella* has been used by only two authors (*Peck, 1937a*; *Peck, 1937b*; *Pratt, 1939*). *Townes (1939)* used the general term apical lobe of the *volsella*, while *Quicke & van Achterberg (1990)*, following *Tobias (1967)*, used *cuspidal processes*, but these terms have never been used outside these works.

***Preferred term.*** *Cuspis.*

***Morphological variation in Ichneumonoidea.*** The presence of the *cuspis* varies within the different subfamilies. In fact, it is clearly present in Labeninae (Fig. 8D), Rhyssinae (Fig. 6) and Xoridinae (Fig. 5A) but is indistinguishable in *Melanichneumon* (Ichneumoninae) (Fig. 1C). *Pratt (1939)* also found differences among Acaenitini (*cuspis* absent) and Xoridinae (*cuspis* present). We confirmed the presence of a *cuspis* in Xoridinae.

From our dissections, the *cuspis* is absent in *Netelia* (Figs. 11A–11B), but *Delrio* (*1975*, p. 9), stated that ''*les parties apicoventrales (section 3) sont bifurqueés et forment la volsella, avec un digitus médian et une cuspis latérale* (=*the apicoventral parts (section 3) are bifurcated and form the volsella, with a median digitus and a lateral cuspis*)'', which seems to provide evidence for the presence of a *cuspis* in *Netelia*. However, it is unclear if Delrio refers to the entire *parossiculus*, to its apical region (=the actual *cuspis*), or to the *gonossiculus*, and therefore misidentified the sclerites. Further studies are required to fully understand the the variation of this area in *Netelia*.

**IV.2. GONOSSICULUS** (gss, Figs. 6A, 6C, 7A, 8B)

*squama* by *Thomson (1872)*.

*valva interna* by *Zander (1900)*.

*lacinia* by *Wheeler (1910)*.

*gonossiculus* by *Crampton (1919)*; *Schulmeister (2001)*; *Schulmeister (2003)*; *Mikó et al. (2013)*.

*copulatory ossicle* by *Crampton (1919)*.

*pièce en trébuchet* by *Boulangé (1924)*.

*sagitta* by *Ross (1937)*; *Pratt (1939)*; *Townes (1939)*.

*gonolacinia* by *Peck (1937a)*; *Peck (1937b)*.

*digitus* by *Snodgrass (1941)*; *Michener (1944)*; *Alam (1952)*; *Michener (1956)*; *Snodgrass (1957)*; *Delrio (1975)*; *Tremblay (1979)*; *Birket-Smith (1981)*; *Tremblay (1981)*; *Tremblay (1983)*; *Johnson (1984)*; *Maetô (1996)*; *Schulmeister (2001)*; *Schulmeister (2003)*; *Belokobylskij, Zaldivar-Riveron & Quicke (2004)*; *Konishi (2005)*; *Brajković et al. (2010)*; *Žikić et al. (2011)*; *Karlsson & Ronquist (2012)*.

*digitus volsellaris* by *Snodgrass (1941)*.

*inner paramere* by *Arnold (1951)*.

*forcipes exteriores* by *Haupt (1962)*.

*parameres interiores* by *Priesner (1966)*.

*\*lateropenite* by *Boudinot (2018)*.

**Concept.** The *gonossiculus* is the distal sclerite of the *volsella*, articulated with the *parossiculus*. It consists of two areas: (1) the *basiura* (bsr, Figs. 6A), which is the proximal part of the sclerite and is the site of insertion of the *parossiculo-gonossicular muscles*, and (2) the *apiceps* (aps, Fig. 6B), which is the distal part of the sclerite. Despite the many different terms applied to this sclerite, there has never been confusion regarding its identification.

**Definition.** As defined by the HAO, the *gonossiculus* is the sclerite that is located on the distoventral part of the *gonostyle/volsella complex*, and is articulated with the more proximal sclerites of the *gonostyle/volsella complex* (Table 2).

**Discussion of terminology.** At least 14 terms have been introduced to refer to the *gonossiculus*. Of these, four (*gonossiculus, sagitta, gonolacinia, digitus*) are worth discussing as they have been employed more than once after their introduction. All of the other terms (see above) are rejected as they do not conform to criterion 2.

*Crampton (1919)* introduced the term *gonossiculus* to refer to a distal sclerite articulated with the *parossiculus*, while *Ross (1937)* introduced the term *sagitta* for the same sclerite. However, *Peck (1937a)* noticed that this latter term was already in use for the *penisvalva* in *Bombus* (Apidae), and he replaced it with *gonolacinia* (following criterion 4). Later on, *Snodgrass (1941)* introduced the term *digitus*, which became popular among hymenopterists.

*Schulmeister (2001)*, surveying the terminology, rejected *sagitta*, for the same reasons advanced by *Peck (1937a)*, and *gonolacinia*, since the term had not been employed after *Peck (1937a)*; *Peck (1937b)* (rejected by criterion 2). However, *Schulmeister (2001)* could not decide whether to use *gonossiculus* and *digitus*, and she opted to use both of them interchangeably. This latter action, however, did not help build a unified terminology. In

fact, different authors, preferred different terms. For example, *Mikó et al. (2013)* opted for *gonossiculus*, while *Boudinot (2013)* chose *digitus*.

Choosing between *gonossiculus* and *digitus* is not easy. If we consider the oldest name, then *gonossiculus* should be the preferred term. Still, if we consider common usage, then the preferred term would be *digitus*. However, it must be noted that *digitus* is also applied to a structure in Lepidoptera, which is not homologous to the *digitus* in Hymenoptera (*Tuxen, 1970*), and potentially it has been used in other insect orders as well, even though not reported in the glossary. Therefore, to avoid future confusion, we recommend the use of *gonossiculus* because it is the oldest name (fulfilling criterion 3), and the one that does not correspond to any other structure in other orders (fulfilling criterion 4).

The term *gonossiculus* has never been used in Ichneumonoidea, preferring the synonym *digitus* (*e.g.*, *Konishi, 2005*; *Karlsson & Ronquist, 2012*).

***Preferred terms.*** *Gonossiculus.*

***Morphological variation in Ichneumonoidea.*** The major feature of the *gonossiculus* seems to be its presence or absence. In fact, Ichneumonidae shows variability in the sclerite, which is present and articulated in *Labena grallator* (Labeninae) (Figs. 8B, 9A–9B), Poemeninae (Fig. 7), Rhyssinae (Figs. 6A & 6C), Tryphoninae (Figs. 11A–11B), and Xoridinae (Fig. 5A) but absent in *Melanichneumon lissorufus* (Ichneumoninae) (Figs. 1C) and Mesochorinae (Fig. 4A). The length of the *gonossiculus* is another important character. *Pratt (1939)* realized that its length varies according to the different taxa (*e.g.*, shorter in Ichneumoninae and longer in Pimplinae, Labeninae, and some Tryphoninae). This is confirmed in this paper as well: *Labena grallator* (Labeninae) (Figs. 8B, 9A–9B), *Neoxorides* (Poemeninae) (Fig. 7), *Rhyssa* (Rhyssinae) (Figs. 6A, 6C) and *Xorides* (Xoridinae) (Fig. 5A) have a very elongated *gonossiculus*, while in *Netelia* (Tryphoninae) (Fig. 11A–11B) and Mesochorinae (Fig. 4A) the *gonossiculus* is extremely reduced . In Braconidae, the literature is unclear. However, according to *Quicke & van Achterberg (1990)*, there is a range of variability depending on the subfamily.

**IV.2.a. BASIURA** (bsr, Fig. 9A)
*basiura* by *Ross (1945)*.
*digiura* by *Schulmeister (2001)*; *Schulmeister (2003)*.

***Concept.*** The *basiura* is the proximal area of the *gonossiculus*. There has never been much confusion regarding the identification of this area, even though its unclear boundaries led many authors to not discuss it in their work (*e.g.*, *Karlsson & Ronquist, 2012*).

***Definition.*** As defined by the HAO, the *basiura* is the area that is the proximal part of the *gonossiculus* and corresponds to the insertion of the *medial penisvalvo-gonossicular muscle* (Table 2).

***Discussion of terminology.*** The term *basiura* was introduced by *Ross (1945)* to identify the basal portion of the *gonossiculus*. After that, no other terms were introduced for the same area, with the exception of *digiura* by *Schulmeister (2001)*, who did so because she realized that *Ross (1945)* did not clearly define the limit of this area.

However, due to the paucity of the usage of these two terms (as far as we know, there are no authors who identified the different areas of the *gonossiculus* except for the two above and never outside the sawflies), we can base the decision of the preferred term based solely on the date of introduction (following criterion 3). Therefore, the preferred name should be *basiura*.

The term *basiura* has never been used in Ichneumonoidea.

***Preferred terms.*** *Basiura*.

***Morphological variation in Ichneumonoidea.*** It is extremely difficult to observe the delimitation of different areas of the *gonossiculus* in Ichneumonoidea. Several previous researchers did not discuss or identify the two areas (*e.g.*, *Peck, 1937a*; *Peck, 1937b*; *Pratt, 1939*; *Karlsson & Ronquist, 2012*).

**IV.2.b. APICEPS** (aps, Fig. 9B)
*apiceps* by *Ross (1945)*.
*digiceps* by *Schulmeister (2001)*; *Schulmeister (2003)*.

***Concept.*** The *apiceps* is the distal area of the *gonossiculus*. There has never been much confusion regarding the identification of this area, even though its unclear boundaries led many authors to not discuss it in their work (*e.g.*, *Karlsson & Ronquist, 2012*).

***Definition.*** As defined by the HAO, the *apiceps* is the area that is the distal part of the *gonossiculus* and is connected to the *parossiculus* via membranous conjunctiva (Table 2).

***Discussion of terminology.*** The term *apiceps* was introduced by *Ross (1945)* to identify the basal portion of the *gonossiculus*. After that, no other terms were introduced for the same area, with the exception of *digiceps* by *Schulmeister (2001)*, who did so because she realized that *Ross (1945)* did not clearly define the limit of this area.

However, due to the paucity of the usage of any of these two terms (as far as we know there are no authors who identified the different areas of the *gonossiculus* except for the two above, and never outside the sawflies), we can base the decision of the preferred term based solely on the date of introduction (following criterion 3). Therefore, the preferred name should be *apiceps*.

The term apiceps has never been used in Ichneumonoidea.

***Preferred terms.*** *Apiceps*.

***Morphological variation in Ichneumonoidea.*** It is extremely difficult to observe the delimitation of different areas of the *gonossiculus* in Ichneumonoidea. Several previous researchers (*e.g.*, *Peck, 1937a*; *Peck, 1937b*; *Pratt, 1939*; *Karlsson & Ronquist, 2012*) did not discuss or identify the two areas.

***Comments on volsella and its associated elements.*** *Vilhelmsen (1997)* identified the entire *volsella* as a hymenopteran synapomorphy. According to *Boudinot (2018)*, only the *parossiculus* is an actual synapomorphy among Hymenoptera, not the entire *volsella*. This reassessment was based on the fact that only the *parossiculus* bears intrinsic musculature,

absent in other orders. Further studies are required to corroborate *Boudinot*'s (*2018*) hypothesis.

## V. PENISVALVA

 **V. PENISVALVA**  (pv, Figs. 1D, 2, 3A–3B, 4, 5, 6A–6B, 7, 9C–9D, 10A, 10C–10D, 11)
*thyrses* by *Audouin (1821)*.
*baguette du fourreau* by *Dufour (1841)*.
*sagitta* by *Thomson (1872)*.
*crochet* by *Seurat (1898)*.
*valve du pénis* by *Boulangé (1914)*.
*penisvalva* by *Crampton (1919)*; *Townes (1939)*; *Snodgrass (1941)*; *Michener (1944)*;
    *Alam (1952)*; *Michener (1956)*; *Snodgrass (1957)*; *Olmi (1984a)*; *Olmi (1984b)*;
    *Schulmeister (2001)*; *Schulmeister (2003)*; *Žikić et al. (2011)*; *Karlsson & Ronquist (2012)*;
    *Boudinot (2013)*; *Mikó et al. (2013)*.
*penis rod* by *Crampton (1919)*.
*penis valve* by *Crampton (1919)*; *Snodgrass (1941)*; *Ross (1945)*; *Johnson (1984)*.
*valve composing penis* by *Crampton (1919)*.
*paramere* (part.) by *Beck (1933)*; *Peck (1937a)*; *Peck (1937b)*.
*lamina aedeagalis* by *Snodgrass (1941)*.
*aedeagal sclerite* by *Snodgrass (1941)*.
*aedeagus* (part.) by *Alam (1952)*; *Peck (1937a)*; *Peck (1937b)*; *Pratt (1939)*; *Snodgrass (1941)*;
    *Snodgrass (1957)*; *Wahl & Gauld (1998)*; *Konishi (2005)*; *Watanabe & Matsumoto (2010)*;
    *Žikić et al. (2011)*; *Watanabe, Taniwaki & Kasparyan (2015)*; *Broad, Shaw & Fitton (2018)*;
    *Bennett et al. (2019)*; *Brajković et al. (2010)*.
*\*edeago* by *Tremblay (1979)*; *Tremblay (1981)*; *Tremblay (1983)*.
*inner or dorsal arm of the stipes* by *Arnold (1951)*.
*aedeagal rod* by *Snodgrass (1957)*.
*thyrsos* by *Birket-Smith (1981)*.
*\*valva* by *Brajković et al. (2010)*.
*\*penial sclerite* by *Boudinot (2018)*.
*\*penis* by *Boudinot (2018)*.

 **Concept.** The *penisvalva* is a paired sclerite located in the middle of the external male genitalia. Each of the two *penisvalvae* consists of two areas: (1) the *valviceps* (vvc, Fig. 9C) located distally; and (2) the *valvura* (vvr, Fig. 9D), located proximally. Where the two areas meet, there is the *ergot*, an apodeme which is the site of insertion of the *lateral* (gs-pvl) and *distoventral gonostyle/volsella complex-penisvalval muscle* (gs-pvdv) (Figs. 3B–3C, 4B & 5A–5B), and the *parossiculo-penisvalval muscle* (pss-pv).

 The majority of the misconceptions regarding this sclerite are based on two terms: *aedeagus* and *penisvalva* (see below).

 **Definition.** As defined by the HAO, the *penisvalva* is the sclerite that is in the middle of the external male genitalia, and surrounds the distal part of the *ductus ejaculatorius* and the *endophallus* (Table 2).

***Discussion of terminology.*** There has been little confusion regarding the term *penisvalva*. It was introduced by *Crampton (1919)*, referring to the two sclerites in the middle of the male external genitalia, and only *Birket-Smith (1981)* tried to replace the term with *thyrso*, based on *thyrses* (*Audouin, 1821*). However, this latter term was not applied any further within Hymenoptera, preferring the simpler *penisvalva*. Only *Boudinot (2018)* decided to use the term *penial sclerite*.

As already mentioned above, the majority of the problems involve the terms *aedeagus* and *penisvalva*. In fact, the former has been used interchangeably with *penisvalva* (*e.g.*, *Brajković et al., 2010*; *Žikić et al., 2011*; *Broad, Shaw & Fitton, 2018*; *Bennett et al., 2019*). However, despite being regularly used in recent years, *aedeagus* is correctly applied to the *penisvalvae+endophallus*, and not just to one sclerite. Therefore, *penisvalva*, which is restricted to Hymenoptera, is the preferred term over *aedeagus* when referring just to the sclerites, while *aedeagus* is the preferred term when used to refer to both the two *penisvalvae* and the *endophallus*.

Except for *Karlsson & Ronquist (2012)*, all the recent works on the external genitalia in Ichneumonoidea refer to the two *penisvalvae* as the *aedeagus* (*Peck, 1937a*; *Peck, 1937b*; *Pratt, 1939*; *Alam, 1952*; *Konishi, 2005*; *Žikić et al., 2011*; *Broad, Shaw & Fitton, 2018*; *Bennett et al., 2019*; *Brajković et al., 2010*).

***Preferred term.*** *Penisvalva*.

***Morphological variation in Ichneumonoidea.*** Within Ichneumonoidea, there seems to be variability of the *penisvalva*. In *Labena grallator*, the separation of the *penisvalvae* is not discernible dorsomedially, and the *penisvalvae* are dorsoventrally flattened (Figs. 2A–2C), preventing, or at least reducing, the ability of the *penisvalvae* to open. This condition also seems to be present in Ichneumoninae (Fig. 1D), while *Netelia* (Tryphoninae) and *Temelucha* (Cremastinae) appear to have the two *penisvalvae* completely divided (Fig. 3A, 10A). In the latter taxa, the *penisvalvae* seems not to be separated ventromedially from the *gonostyle* (Fig. 3A). The size of the *penisvalva* also varies. In *Netelia* (Figs. 10C–10D) and *Mesochorus* (Figs. 4A–4C) the structure is roughly half the length of the *gonostyle*, while in all the other Ichneumonidae (*e.g.*, Figs. 3, 6), it is as long as the *gonostyle*. *Karlsson & Ronquist (2012)* observed that in both the species of Braconidae studied by them, the *penisvalva* is longer than the *volsella* and the *gonostyle*.

**V.a. VALVURA**  (vvr, Figs. 7B, 9D)

*apophyse péniale* by *Boulangé (1924)*.

*valvura* by *Ross (1945)*; *Schulmeister (2001)*; *Schulmeister (2003)*; *Karlsson & Ronquist (2012)*; *Boudinot (2013)*.

*aedeagal apodeme* by *Snodgrass (1941)*; *Alam (1952)*; *Snodgrass (1957)*; *Konishi (2005)*; *Brajković et al. (2010)*.

*apodeme of penis valve* by *Michener (1965)*.

\**apodema edeagale* by *Tremblay (1979)*; *Tremblay (1981)*; *Tremblay (1983)*.

*apodema thyrsos* by *Birket-Smith (1981)*.

\**penial apodeme* by *Boudinot (2018)*.

*Concept.* The *valvura* is the proximally located area of the *penisvalva*. Other than the original, no other concepts have been applied to this area.

*Definition.* As defined by the HAO, the *valvura* is the area that is located proximally to the *ergot* on the *penisvalva* (Table 2).

*Discussion of terminology.* Only two terms have been employed by more than one author since their introduction: *valvura* and *aedeagal apodeme*.

The term *valvura* was introduced by *Ross (1945)* and was applied to the long basal structure of the *penisvalva*. Later on, *aedeagal apodeme* was introduced by *Snodgrass (1941)* to refer to the same area. However, *aedeagal apodeme* is also used for a non-homologous structure in Siphonaptera (*Tuxen, 1970*). Therefore, *valvura* is not only the first term to be introduced (fulfilling criterion 3), but it is also the name that does not correspond to any other structure in other orders (fulfilling criterion 4). *Valvura* should be considered the preferred term.

Except for *Karlsson & Ronquist (2012)*, who applied the term valvura to the basal part of the *penisvalva*, the other authors either did not mention the term or they used *aedeagal apodeme* (*Alam, 1952*; *Konishi, 2005*; *Brajković et al., 2010*).

*Preferred term.* Valvura.

*Morphological variation in Ichneumonoidea.* The overall morphology of the *valvura* in Ichneumonoidea, seems to vary according to certain taxa. *Peck (1937a)* observed that the *valvura* of some Anomaloninae (*Agrypon flaveolatum* (Gravenhorst, 1807) and *Therion circumflexum* (Linnaeus, 1758)) is visibly medio-laterally expanded (Figs. 129, 141 *in Peck, 1937a*, p. 251), as it is in many Banchinae (*Pratt, 1939*). Also, the shape of the *ergot* varies significantly, being extremely produced in Pimplinae (*e.g.*, *Dolichomitus tuberculatus* (Geoffrey, 1875)) (Figs. 147, 141 *in Peck, 1937a*, p. 252) but extremely reduced in *Megharyssa macrura* (Figs. 134, 141 *in Peck, 1937a*, p. 251). In *Temelucha* (Cremastinae), there is also a well-defined, pointed *ergot* (erg, Fig. 3C).

 **V.b. VALVICEPS**  (vvc, Fig. 9C)
*valviceps* by *Ross (1945)*; *Schulmeister (2001)*; *Schulmeister (2003)*.
\**penisvalva* (part.) by *Konishi (2005)*.
\**distal portion* by *Boudinot (2018)*.

*Concept.* The *valviceps* is the proximally located area of the *penisvalva*. Other than the original, no other concepts have been applied to this area.

*Definition.* As defined by the HAO, the *valviceps* is the area that is the distal partof the *penisvalva*, distally of the *ergot* (Table 2).

*Discussion of terminology.* The term *valviceps* was introduced by *Ross (1945)* and was applied to the apical part of the *penisvalva*. No other term has been introduced to refer to this area, with the exception of *Konishi (2005)*, who applied the term *penisvalva* for this area (instead of referring to the entire sclerite). This makes *penisvalva* a case of homonymy.

The term *valviceps* has never been used in Ichneumonoidea.

*Preferred term. Valviceps.*

*Morphological variation in Ichneumonoidea.* Very little information is available for the *valviceps* in Ichneumonoidea. From what can be observed in *Peck*'s (*1937a*, p. 251–252) drawings, the majority of the variability for this area relies on the overall curvature of the part. For instance, *Virgichneumon maculicauda* (Perkins, 1953) (Ichneumoninae), in lateral view, has a distinct sinuosity of the *valviceps*, possibly functioning as muscle attachments (Fig. 135 *in Peck, 1937a*, p. 251). More exploration is needed.

*Comments on penisvalva and its associated elements. Boudinot (2018)* proposed the term *penial sclerite* (for one sclerite) and *penis* (for both sclerites), believing that the *penisvalva* of the Hexapoda derived from the medially fused primary gonopods (gonopore-bearing limbs).

## VI. MEDIAN SCLEROTIZED STYLE

 **VI. MEDIAN SCLEROTIZED STYLE**  (Figs. 8A, 9A *in Schulmeister, 2003*, p. 261–262)
*median sclerotized style* by *Ross (1937)*; *Schulmeister (2001)*; *Schulmeister (2003)*.
*ventral rod of aedeagus* by *Snodgrass (1941)*.
*detached rhachies* by *Smith (1969)*; *Smith (1970a)*; *Smith (1970b)*.

*Concept.* The *median sclerotized style* is a long, thin sclerite located between the two *penisvalvae*. One muscle, the *penisvalvo-median sclerotized style muscle*, connects the *valvura* to the *median sclerotized style*.

*Definition.* As defined by the HAO, the *median sclerotized style* is the sclerite that is located ventrally between the *penisvalvae* (Table 2).

*Discussion of terminology.* The term *median sclerotized style* was introduced by *Ross (1937)* to identify a sclerite present in Siricidae and Cephidae. Later on, *Smith (1969)* introduced *detached rhachies* for the same structure, while *Snodgrass (1941)* used the term

*ventral rod of aedeagus.* This latter term has not been used after its introduction and can be rejected following criterion 2. There are no major differences in the use of *median sclerotized style* and *detached rhachies* as both were employed in three papers. Therefore the preferred between the two is *median sclerotized style* because it was the first to be introduced (fulfilling criterion 3).

The term *median scelrotized style* has never been used in Ichneumonoidea.

**Preferred term.** *Median sclerotized style.*

**Morphological variation in Ichneumonoidea.** A sclerite located between the two *penisvalvae* has never been located in Ichneumonoidea (*e.g.*, *Schulmeister, 2003*).

**Comments on the median sclerotized style.** According to *Schulmeister (2001)*, the *median sclerotized style* has been recorded only in Cephidae and Siricidae, and is located in the same place where the *phallotrema* is located in other hymenopteran taxa (*e.g.*, Ichneumonoidea). According to *Smith (1970a)*, the *median sclerotized style* is a sclerite originally separated from the *penisvalvae.*

In Apoidea, other types of sclerotization of the membrane are present, but are not homologous to the *median sclerotized style* (*Schulmeister, 2001*). One example is the *spatha* which is an unpaired sclerite located dorsally of the basal section of the *aedeagus* of some Aculeata.

**VII. FIBULA DUCTI**

 **VII. FIBULA DUCTI**  (Fig. 3B *in Schulmeister, 2003*, p. 256)
\**sperrkeil* by *Clausen (1938)*.
\**wedge* by *Forbes & Do-Van-Quy (1965)*.
*fibula ducti* by *Schulmeister (2001)*; *Schulmeister (2003)*.
\**wedge sclerite* by *Boudinot (2013)*.

**Concept.** The *fibula ducti* is a small sclerite located in the proximal section of the *ductus ejaculatorius.* It is considered part of the internal male genitalia.

**Definition.** As defined by the HAO, the *fibula ducti* is the sclerite that is located in the proximal end of the unpaired part of the *ductus ejaculatorius* (Table 2).

**Discussion of terminology.** The first author to identify a sclerotized area on the *ductus ejaculatorius* was *Clausen (1938)* who coined the term *sperrkeil* in *Formica rufa* Linnaeus, 1761 (Formicidae). Later on, *Forbes & Do-Van-Quy (1965)* identified a similar structure in *Neavamyrmex* Borgmeier, 1940 (Formicidae), referring to it with the general term *wedge*, while (*Smith, 1990*, p. 46) depicted a sclerite in the same position in Pergidae without naming it. *Schulmeister (2001)* acknowledged *Smith*'s (*1990*) image, and decided to name the sclerite *fibula ducti*. More recently, *Boudinot (2013)*, while referring to the *wedge* of *Forbes & Do-Van-Quy (1965)*, named the same structure *wedge sclerite.*

Among these terms, only one best fulfills criteria 2 and 4, and therefore should be considered the preferred term: *fibula ducti*. In fact, the others have not been used after the

original introduction (*e.g.*, *sperrkeil*) or generally refer to the shape of the sclerite rather than provide a specific term for it (*e.g.*, *wedge*).

> **Preferred term.** *Fibula ducti.*

> **Morphological variation in Ichneumonoidea.** A sclerite located proximally on the *ductus ejaculatorius* has never been located in Ichneumonoidea (*e.g.*, *Schulmeister, 2003*).

**Comments on the fibula ducti.** According to *Schulmeister (2001)*, more investigation is needed to understand if the *sperrkeil* of *Clausen (1938)* is homologous to the *fibula ducti*. However, the similar location (proximal to the *ductus ejaculatorius*) and the similar shape are good indicators that the two are very likely the same structure and thus are synonymized here.

So far, the *fibula ducti* was identified only in three families of Hymenoptera: Formicidae (*Clausen, 1938*; *Forbes & Do-Van-Quy, 1965*; *Boudinot, 2013*), Pergidae (*Smith, 1990*; *Schulmeister, 2001*), and Argidae (*Schulmeister, 2003*).

## The *Netelia* case

As mentioned above, the external male genitalia of Ichneumonoidea have been very little explored. The exception is the genus, *Netelia* Gray, belonging to the subfamily Tryphoninae (Ichneumonidae). With more than 330 valid species worldwide (*Yu, van Achterberg & Horstmann, 2016*), the genus *Netelia* has a long tradition of male genitalia description, so much that some terms have been coined and used only within the genus.

*Pad* and *brace* refer to structures that are difficult to interpret (Figs. 10C–10D, 11). Both are located in the inner part of the *gonostyle* and are continuous with each other (Fig. 10D). *Townes (1939)* defined *brace* as a sclerome located basally to the *pad*, which is a "*flap or vescicle that sticks out*". Since then, these two terms have been employed and are still in use (*e.g.*, *Broad, 2012*; *Konishi, Chen & Pham, 2022*). It is very difficult to understand what those two structures are homologous with since these two terms have not been utilized in other insect groups (*Tuxen, 1956*; *Tuxen, 1970*), but we draw some possible conclusions.

The two *pads* are located medially, between the two *gonostyles* and distal to the two *penisvalvae*. They are connected between each other along their proximo-medial internal margin, and are in continuity with a conjunctiva for the entire length of their external margins. This conjunctiva covers the internal distal portion of the *gonostyle*, and is delimited proximally by the *brace*, a more sclerotized area that runs proximo-distally and that is in connection with the *pad* in its distal part (Figs. 10C–10D). Being composed of conjunctiva, the two *pads* are flexible, and once they are disconnected from each other, they fold towards the internal part of the *gonostyle* (Figs. 10C, 11A–11B). The proximal section of this entire structure (*brace* + *conjunctiva* + *pad*) is the site of insertion of the tendon belonging to a muscle that inserts into the proximal margin of the *gonostyle* (Fig. 11).

Within Hymenoptera, only one muscle shares the same insertion of this muscle as in *Netelia*: the *proximal gonostyle/volsella complex-harpal muscle* (muscle t' of *Schulmeister (2001)*; *Schulmeister (2003)*). This muscle, together with the *distal gonostyle/volsella complex-harpal muscle* (muscle t" of *Schulmeister (2001)*; *Schulmeister (2003)*), is tightly

linked to the presence of the *harpe* (*Schulmeister, 2003*). In Ichneumonoidea, the *harpe* is not discernible as a distinct sclerite, nor is it in *Netelia*, and thus, the *proximal* and *distal gonostyle/volsella complex-harpal muscle* should be absent. However, their presence (even though not recognized as two different muscles) was discovered in *Megarhyssa lunator* by *Peck (1937a)*, who associated them with the presence of a *harpe* that was only partially distinct ("incompletely fused" for the author). *Mikó et al. (2013)* also reported the same muscle in Ceraphronoidea without a *harpe* and proposed a change of muscle function for the *proximal gonostyle/volsella complex-harpal muscle* that went from moving the *harpe* to laterally bending the *gonostyle*. In *Netelia* the *proximal gonostyle/volsella complex-harpal muscles* (gs-hrp, Fig. 11A) could also have changed its function, and it is used to bend the entire complex formed by the *brace*, the *pad* and the rest of the conjunctiva.

Regarding the conjunctiva, it is not clear what it might be homologous with. *Schulmeister (2001)*, reported the presence of the conjunctiva located on the distal portion of the *harpe* in certain families of basal Hymenoptera. The structure was firstly called *gonomacula* by *Crampton (1919)* and several authors advanced the hypothesis that the *gonomacula* could serve as a suction cup to hold the female during copulation, due to the presence of a muscle, the *harpo-gonomacular muscle* (ha-gon, Table 3). Even though the conjunctiva and the pad in *Netelia* share the same composition (conjunctiva) and a similar position of the *gonomacula*, we refrain from treating them as homologous because a muscle in the same position of the *harpo-gonomaculal muscle* has not been retrieved.

However, it is possible that the conjunctiva in *Netelia* evolved independently from the *gonomacula* in basal Hymenoptera to perform a similar function. In fact, the *penisvalva* in *Netelia* is extremely short, barely reaching half of the length of the *gonostyle*. Therefore, the conjunctiva could help in holding the female during copulation (functioning as a suction cup), while the *pad* helps the transfer of the sperm (functioning as a receptacle) to the female genital organs. Further comparative analyses including members of the subfamily Tryphoninae and a broader representation of Ichneumonidae will be needed to fully understand the function and identity of these structures. Investigations on the internal copulatory organ of females will likely be informative.

## Musculature of Ichneumonoidea male genitalia

Musculature is essential for a comparative framework to understand relations among skeletal features (*e.g.*, *Vilhelmsen, 2010*; *Mikó et al., 2012*). Even though a thorough study of the musculature of the male genitalia in Ichneumonoidea is beyond the scope of the current study, we are providing the following resources to facilitate future researchers in this field: (1) definitions and abbreviations of the different muscles according to an ontological framework (Table 3); (2) an alignment of the terminology for the musculature of the entire order Hymenoptera (Table 4); (3) a summary of the observations and analyses of the musculature of Ichneumonoidea (Table 5).

The musculature of the Ichneumonoidea male genitalia remains mostly unexplored, with only a handful of authors studying a very limited number of species. The first to fully analyze the muscles and their possible functions in Ichneumonoidea were *Peck (1937a)* for *Megarhyssa lunator* (Rhyssinae), and *Alam (1952)* for *Stenobracon deesae* (Braconinae).

Table 5 Muscle presence or absence per family within Ichneumonoidea, according to different authors. 0, absent; 1, present; –, missing data.

| Abbreviation | Muscle | Ichneumonidae (*Schulmeister, 2003*) | Ichneumonidae (*Peck, 1937b*) | Braconidae (*Alam, 1952*) | Confirmed in this paper |
|---|---|---|---|---|---|
| S9-cm | Medial S9-cupulal muscle | 0 | 1 | 1 | 1 |
| S9-cml | Mediolateral S9-cupulal muscle | 1 | 1 | 1 | 1 |
| S9-cl | Lateral S9-cupulal muscle | 1 | 1 | 1 | 1 |
| c-gsvm | Ventromedial cupulo-gonostyle/volsella complex muscle | 1 | 1 | 1 | 1 |
| c-gsvl | Ventrolateral cupulo-gonostyle/volsella complex muscle | 1 | 1 | 1 | 1 |
| c-gsdl | Dorsolateral cupulo-gonostyle/volsella complex | 1 | 1 | – | 1 |
| c-gsdm | Dorsomedial cupulo-gonostyle/volsella complex muscle | 1 | 1 | 1 | 1 |
| gs-pvpv | Proximoventral gonostyle/volsella complex - penisvalval muscle | 1 | 1 | 1 | 1 |
| gs-pvdv | Distoventral gonostyle/volsella complex - penisvalval muscle | 1 | 1 | 1 | 1 |
| gs-pvdd | Distodorsal gonostyle/volsella complex - penisvalval muscles | 1 | 1 | 1 | 1 |
| gs-pvpd | Proximodorsal gonostyle/volsella complex - penisvalval muscle | 1 | 1 | – | 1 |
| gs-pvl | Lateral gonostyle/volsella complex-penisvalval muscle | 0 | 1 | 1 | 1 |
| pv-gssl | Lateral penisvalvo-gonossiculal muscle | 0 | 1 | – | 1 |
| pv-gssm | Medial penisvalvo-gonossiculal muscle | 0 | 1 | 1 | 1 |
| pv-ph | Penisvalvo-phallotremal muscle | 0 | 1 | 1 | 1 |
| gss-ph | Gonossiculo-phallotremal muscle | 0 | 1 | 1 | – |
| pss-ph | Parossiculo-phallotremal muscle | 1 | 1 | 1 | – |
| gs-pss | Gonostyle/volsella complex-parossicular muscle | 1 | 1 | 1 | 1 |
| gn-pssp | Proximal gonostipo-parossicular muscle | – | – | 1 | 1 |
| gn-pssd | Distal gonostipo-parossicular muscle | – | – | – | 1 |
| imvll | Lateral gonostyle/volsella complex-volsella muscle | 1 | 1 | 1 | 1 |
| imvl | Median gonostyle/volsella complex-volsella muscle | 1 | 1 | 1 | 1 |
| imvm | Gonostyle/volsella complex-gonossiculus muscle | 0 | 1 | 1 | 1 |
| pss-pv | Parossiculo-penisvalval muscle | 1 | – | – | 1 |

*(continued on next page)*

**Table 5** (*continued*)

| Abbreviation | Muscle | Ichneumonidae (*Schulmeister, 2003*) | Ichneumonidae (*Peck, 1937b*) | Braconidae (*Alam, 1952*) | Confirmed in this paper |
|---|---|---|---|---|---|
| gs-hrd | Distal gonostyle/volsella complex-harpal muscle | 0 | 1 | – | 0 |
| gs-hrp | Proximal gonostyle/volsella complex-harpal muscle | 0 | 1 | – | 1 (only in *Netelia*) |
| gs-hra | Apical gonostyle/volsella complex - harpal muscles | 0 | 0 | – | 0 |
| ha-gon | harpo-gonomacular muscle | 0 | 0 | – | 0 |
| gs-gs | intragonostyle muscle | 0 | – | – | – |
| pv-pv | interpenisvalval muscle | 1 | – | – | 1 |
| vl-vl | intervolsellal muscle | – | – | – | – |
| pv-mss | penisvalvo-median sclerotized style muscle | 0 | – | – | 0 |

While *Snodgrass (1941)* also provided some comments on the musculature of the male genitalia, his assessment was not comprehensive and focused only on the muscles of the *volsella*. The most recent study on the musculature for the superfamily was provided by *Schulmeister (2003)*, who studied two specimens, generally referred to as "Ichneumonidae sp.1" and "Ichneumonidae sp.2." By comparing the results of these studies (Table 5), we can see some inconsistencies. *Peck (1937a)* identified 20 muscles, *Alam (1952)* only 18, and *Schulmeister (2003)* only 16. It is unclear if this difference is due to variability within and across the families or simply a different interpretation of muscles between authors. In the current studies we have found 24 muscles, confirming all the muscles already reported by previous authors.

There is inconsistency in the presence or absence of certain muscles (see Table 5). According to *Schulmeister (2003)*, only two of the three muscles connecting the *cupula* to the *abdominal sternum 9* are present in Ichneumonidae, namely the *mediolateral* and the *lateral S9-cupulal muscles* (muscle b and c of *Schulmeister (2001)*; *Schulmeister (2003)*) (Table 4), but *Peck (1937a)* claimed to have retrieved the *medial S9-cupulal muscle* (muscle a of *Schulmeister (2001)*; *Schulmeister (2003)*) in the subfamily Rhyssinae, and *Alam (1952)* retrieved the same muscles in Braconidae. This is unusual, as *Schulmeister (2003)* did not observe the muscle within the Apocrita, only in the basal Hymenoptera. In this paper, we were able to retrieve the *medial S9-cupulal muscle* (S9-cm) in Cremastinae (Fig. 3A) and Mesochorinae (Figs. Figs. 4A, 4C), confirming *Peck*'s (*1937a*) observations. Moreover, the position of this muscle is very similar to the one reported by *Alam* (*1952*, p. 5) running almost parallel by inserting laterally on the *abdominal sternum 9* and laterally on the *cupula*.

An interesting muscle recorded in Ichneumonidae only by *Schulmeister (2003)* is the *interpenisvalval muscle* (muscle × in *Schulmeister (2001)*; *Schulmeister (2003)*) which was first described by *Boulangé (1924)* in basal Hymenoptera (*Abia lonicerae* (Linnaeus, 1758), Cimbicidae). *Schulmeister (2003)* disproved the presence of the muscle in the genus *Abia* but pointed out that only Ichneumonidae (across the Hymenoptera surveyed) bear the *interpenisvalval muscle*. It is a single muscle running transversely, connecting the two *penisvalvae*, and opening and closing them during copulation. We corroborated the
presence of this muscle within the Ichneumonidae, reporting it within the genus *Temelucha* (Ichneumonidae, Cremastinae) (Fig. 3B), and in the species *Rhyssa persuasoria* (Rhyssinae) (Fig. 6B). This is also the first time that this muscle has been imaged. The position of the *interpenisvalval muscle*, however, seems to vary between the two subfamilies. In fact, in Rhyssinae it is located proximally, connecting the two *valvurae* (as reported also by *Schulmeister (2001)*), while in Cremastinae, is located more apically, connecting the two *valviceps*. The latter position has not been previously reported. A possible explanation for this variation is based on the correlation of the *interpenisvalval muscle* with the level of division of the *penisvalvae*. In Cremastinae, the conjunctiva that connects the two sclerites disappears proximally, allowing only a distal opening of the two *penisvalvae*. In Rhyssinae, the conjunctiva seems to be present across the entire length of the *penisvalvae*, and by being located proximally, the *interpenisvalval muscle*, allows a far greater range of opening of the two sclerites. Future studies are required to assess if this muscle is relegated to Ichneumonidae (and to what extent) or present also in Braconidae and other hymenopteran families.

Another confirmation of a muscle within Ichneumonidae is the *proxima gonostyle/volsella complex-harpal muscle* (gs-hrp, Fig. 11A), reported here in *Netelia* (Tryphoninae) for the first time, and only reported within Ichneumonidae by *Peck (1937b)*. For a complete treatment, please read above, under "The *Netelia* case."

*Alam (1952)* described and imaged a muscle that he named "17" (see Fig. 3A *in Alam, 1952*, p. 625), running antero-laterally from the "median plate" of the *penisvalva* to the *valvura*. The author believed the muscle to be an adductor of the *valvura*. *Alam*'s (*1952*) "median plate" is described as a dorsal extension of the wall surrounding the *aedagus*, which is connected via conjunctiva to the *gonostyle*. As far as we know, "median plate" was employed only by *Alam (1952)*, and muscle "17" was not discussed by subsequent authors. In our interpretation, the "median plate" is either the *ergot* or the entire *valviceps* that in Ichneumonoidea is laterally expanded and proximally produced, while muscle "17", could be related to a similar muscle already identified by *Schulmeister (2003)* in Hymenoptera, muscle "nb". This muscle runs from the *valvura* to the *phallotrema* and *Schulmeister (2003)* interpreted it as a subdivision, occurring in certain taxa, of muscle "n" which, when undivided, runs from the *valvura* to the *basiura*. In our survey, we discovered a paired muscle in *Temelucha* sp. (Cremastinae) that originates from the *valvura* and attaches to a membranous area adjacent to the *ergot*. *Alam (1952)* possibly misinterpreted the insertion of his muscle "17", which actually runs from the *valvura* to the *phallotrema*, and we treat muscle "17" as the *penisvalvo-phallotremal muscle* (pv-ph, Figs. 3B–3C). In Rhyssinae, this muscle is undivided, and only the *medial penisvalvo-gonossicular muscle* (muscle n of *Schulmeister (2001)*) is present (pv-gssm, Fig. 6B).

*Alam (1952)* interpreted muscles "4" and "5" as two different muscles, when they have the same insertion but slightly different attachment (two different, but close, locations, typical of a fan-shaped muscle) and therefore can be considered two bundles of the same muscle (see Fig. 6 *in Alam, 1952*, p. 627).

## CONCLUSION

From the current study, it is clear that the male genitalia in Ichneumonoidea has suffered from terminological inconsistencies that have prevented the exploration of characters in taxonomic studies and establishing correct homology statements that are needed to conduct evolutionary studies. In this regard, the most problematic group of sclerites is the *gonostyle-volsella complex* which has generated several synonyms. Reasons for these synonyms include its overall complexity (*e.g.*, several parts organized differently according to the taxon) and the reduced number of comparative anatomy studies across Hymenoptera (as recently discussed also by *Lanes et al. (2020)* for Bethylidae) that have led different authors to propose new names for the same parts.

Implementation of a larger sample size is required for a more comprehensive anatomical study to advance our understanding of the skeleto-musculature of male genitalia in Ichneumonoidea. This is important for resolving some of the inconsistencies found in the musculature and laying the foundation for testing the phylogenetic signal of male genitalia. The hope is that the tools provided in this contribution will guide and foster further studies on the sclerites and musculature of ichneumonoid male genitalia.

## ACKNOWLEDGEMENTS

A big thank-you goes to Dr. Alicja Copik (University of Central Florida) for allowing the first author (DDP) to use the CLMS at Lake Nona Campus, and to Jeremiah Oyer (University of Central Florida) and Javier Rivera-Huertas (University of Central Florida) who helped DDP access the UCF Lake Nona microscope facility. The first author is also indebted to David Serrano (Broward College) and Miles Y. Zhang (University of Edinburgh) for the specimens received and used in this study. A big thank you goes out to Dr. Kenneth Fedorka and Dr. Eric Goolsby (University of Central Florida) for supporting DDP in his Ph.(D) project, and to Gavin Broad and an anonymous reviewer for substantially improving this manuscript with their comments. Lastly, a big thanks to all the people that, in these many years, supported the first author in his many adventures in the wonderful world of insects. You know who you are.

### Funding

This research was supported by the US National Science Foundation (NSF: DEB-1916788) grant awarded to Barbara J. Sharanowski. Article processing charges were provided by the UCF College of Graduate Studies Open Access Publishing Fund. There was no additional external funding received for this study. The funders had no role in study design, data collection and analysis, decision to publish, or preparation of the manuscript.

### Grant Disclosures

The following grant information was disclosed by the authors:
The US National Science Foundation: NSF: DEB-1916788.

The UCF College of Graduate Studies Open Access Publishing Fund.

## Competing Interests

The authors declare that there are no competing interests.

## Author Contributions

- Davide Dal Pos conceived and designed the experiments, performed the experiments, analyzed the data, prepared figures and/or tables, authored or reviewed drafts of the article, and approved the final draft.
- István Mikó conceived and designed the experiments, performed the experiments, analyzed the data, authored or reviewed drafts of the article, and approved the final draft.
- Elijah J. Talamas analyzed the data, authored or reviewed drafts of the article, and approved the final draft.
- Lars Vilhelmsen analyzed the data, authored or reviewed drafts of the article, and approved the final draft.
- Barbara J. Sharanowski conceived and designed the experiments, analyzed the data, authored or reviewed drafts of the article, and approved the final draft.

## Field Study Permissions

The following information was supplied relating to field study approvals (i.e., approving body and any reference numbers):

Specimens collection was approved by St. Johns River Water Management District.

## Data Deposition

Specimen numbers and deposition information is in Table 1.

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
