# Peer review of "A revised terminology for male genitalia in Hymenoptera (Insecta), with a special emphasis on Ichneumonoidea"

_PeerJ, doi:10.7717/peerj.15874_

## Round 0.1 · original submission · Minor Revisions

Both reviewers agree that the manuscript presents a very good study, awaited and needed by the community. Only minor aspects can be ameliorated based on suggestions of the reviewers, that I encourage you to take into acount before I formally accept the manuscript for publication.

·

Basic reporting

This is a very welcome paper, long overdue. And with these authors, this was clearly going to be an excellent piece of work. As an ichneumonologist, I particularly welcome this work as it should provide the first useful summary of controlled vocabulary and homology for external male genitalia. As I’m about to submit a couple of papers on Netelia taxonomy, this was a particularly timely paper to review! It was really interesting to read about potential homology in the atypical genitalia of Netelia, or that similar structures have evolved again after being lost.

The review of the literature seems comprehensive and there are very sensible rationale for the recommended terms. My only real quibble is that although ‘Ichneumonoidea’ are referred to frequently, the original observations relate to Ichneumonidae, with no Braconidae examined. Therefore, would it not be more appropriate to refer to more frequently to Ichneumonidae while suggesting that the same conclusions apply to Braconidae?

I have a few minor quibbles, a few comments and some questions.

In the Introduction, lines 32-34, I’m not quite sure what is meant by ‘Identical terms describing non-homologous structures (homonyms) span across broadly unrelated taxa, including humans, plants and animals, and even genes’. Do you meant that a term used in humans can be used in plants, other animals [note that humans are animals…] and genes for different structures? Or do you mean that homonyms abound within each of those groups you list?

Lines 59-60: ‘This is true for the hyperdiverse superfamily Ichneumonoidea and its constituent families, Braconidae and Ichenumonidae.’ What exactly is true for Ichneumonoidea? Both of the preceding statements (about missing concepts and the need for updated homology statements)? It’s worth remembering that the HAO was not set up with morphological homology statements as an aim.

Lines 69-70: I get the impression (but I could be completely wrong) that genitalia have not been utilised much at higher taxonomic levels in Hymenoptera because they are not used much as species-specific diagnostic tools. Clearly there are exceptions (e.g., many bees, sawflies, Netelia, Trichogramma) but for most part, there are not easily observable, discrete differences between species, and the male genitalia for the most part are rather simple, compared to moths or many flies. And, of course, the ovipositor, external genitalia, is used a lot.

Lines 187-189: some Japanese taxonomists, particularly Kyohei Watanabe, have made a concerted effort to include male genitalia as part of species descriptions, e.g.:

Watanabe K, Matsumoto R (2010) Disjunctive distribution of the basal genus Aplomerus (Hymenoptera: Ichneumonidae: Xoridinae) in East Asia and North America, with a new species from Japan. Entomological Science 13:375-380
Watanabe K, Tanikawi T, Kasparyan DR (2015) Tanzawana flavomaculata (Hymenoptera, Ichneumonidae, Ctenopelmatinae), a new genus and species of parasitoid of Fagineura crenativora (Tenthredinidae, Nematinae), a serious pest of beech tree. Zootaxa 4040 (2):236-242

Line 224: again, as you mention, I suspect the big reason for male genitalia not being used much in phylogenetic studies is because they are not really used for species-level taxonomy, so they are not routinely dissected and as a consequence many workers are unfamiliar with this bit of the anatomy.

Line 430, and in several other places: you should be referring to Bohart & Menke (not ‘Bohart & S.’).

Line 481: probably irrelevant if you didn’t examine them, but Lusius and some Nematopodius have similar gonostyles to Mesochorinae.

Line 705: typo (‘paraossiculus’)?

Line 781: italicize ‘Melanichneumon’. Also, I thought I could see a cuspis in Netelia. I am probably wrong, but I thought it was a rather weakly sclerotized bit which is pressed against the gonossiculus. I’ve attached a picture I prepared recently, as I was trying to work out the best terms for parts of the Netelia male genitalia. I see I came to some correct conclusions and some incorrect! Delrio (1974) also refers to the cuspis in Netelia.

Lines 779-783: the cuspis seems prominent in many Ctenopelmatinae.

Lines 892-893: ‘Further studies are required to fully understand the extent of such assertions’, is not readily understandable. Am I correct that you are saying further studies are required to test Boudinot’s hypothesis that the parossiculus is a Hymenoptera synapomorphy?

Lines 950-951 and 1112: it seems that in Netelia the gonostyle is greatly enlarged, rather than the penisvalva being smaller than usual. The enlarged external genitalia of Netelia are rather conspicuous. I don’t believe that the pad helps with sperm transfer. I’ve seen Netelia mating (and have some photos, although they aren’t great) and the gonostyles are spread and adpressed against the female’s hypopygium. It looks far more likely to me that the pads are acting as some sort of ‘suction’ devices.

Line 1062: should be ‘more investigation is needed’.

References: again, Bohart and Menke (not ‘Bohart and S.’). Should be just Broad (2012), not Broad, Shaw, Fitton (2012). These keys were so preliminary, Mark and Mike would probably want nothing to do with them anyway! The handbook (Broad et al., 2018) should be referenced as vol. 7, part 12 as pagination is separate in each volume.

Experimental design

no comment

Validity of the findings

See above, under 'basic reporting'.

Reviewer 2 ·

Basic reporting

By reviewing a large literature and selecting the appropriate terms (among a sometimes impressive number of more or less equivalent terms) to describe the skeleto-muscular elements of Hymenoptera male genitalia, the authors intent to standardize this terminology in order to facilitate future taxonomic and evolutionary studies and provide comparability framework.
The text is clear, unambiguous, technically correct. The introduction is sufficient, the background relevant and the literature extensive and appropriate. Tables are very clear and links to HAO portal in fig. 2 is appropriate.
Figures are relevant and appropriately described and labeled but several erroneous references of figures remain in the text. Please check these references carefully.
This manuscript is supposed to give a special emphasis on Ichneumonoidea but the illustrations and the comments actually concern almost only Ichneumonidae. Little information is provided for Braconidae notably in several paragraphs intitled « Morphological variation in Ichneumonoidea » (ex for gonostipes,
parapenis, harpe…) probably because the literature is unclear (as said for other sclerites) or even lacking. Iit would however be interesting to have a better idea of what is known or not in Braconidae compared to Ichneumonidae.

Experimental design

First complete synthetic work about standardization of skeleto-muscular terminology of Hymenoptera male genitalia.
Large literature reviewed.
Comments supported by appropriate illustrations
Selection of adequate skeleto-muscular terms based on predefined criteria whith supporting comments

Validity of the findings

This is a very interesting and important paper which will be very useful to a wide audience within the hymenopterist community because it fills an important gap in the study of male genitalia morphology whose terminology never was standardized, especially in Ichneumonoidea.

Annotated reviews are not available for download in order to protect the identity of reviewers who chose to remain anonymous.

---

## Round 0.2 · accepted · Accept

Thank you for the revised manuscript. I had difficulties reading the tracked-change file (it shows only a few of the changes that you did) but it appears that I can find that all the reviewer's suggestions have been followed.